# Advancements in Genetic and Biochemical Insights: Unraveling the Etiopathogenesis of Neurodegeneration in Parkinson’s Disease

**DOI:** 10.3390/biom14010073

**Published:** 2024-01-05

**Authors:** Yashumati Ratan, Aishwarya Rajput, Ashutosh Pareek, Aaushi Pareek, Vivek Jain, Sonia Sonia, Zeba Farooqui, Ranjeet Kaur, Gurjit Singh

**Affiliations:** 1Department of Pharmacy, Banasthali Vidyapith, Banasthali 304022, Rajasthan, India; kailashaish@gmail.com (A.R.); ashu83aadi@gmail.com (A.P.); aayushipareek26@gmail.com (A.P.); 2Department of Pharmaceutical Sciences, Mohan Lal Sukhadia University, Udaipur 313001, Rajasthan, India; vivek19j@gmail.com; 3Department of Pharmaceutical Sciences, Guru Nanak Dev University, Amritsar 143005, Punjab, India; sonia01pharma@gmail.com; 4Department of Biomedical Engineering, University of Illinois Chicago, Chicago, IL 60607, USA; zeba25@uic.edu; 5Adesh Institute of Dental Sciences and Research, Bathinda 151101, Punjab, India; drranjeetkaur93@gmail.com

**Keywords:** Parkinson’s disease, α-synuclein, molecular mechanism, genetics of Parkinson’s disease, risk factors

## Abstract

Parkinson’s disease (PD) is the second most prevalent neurodegenerative movement disorder worldwide, which is primarily characterized by motor impairments. Even though multiple hypotheses have been proposed over the decades that explain the pathogenesis of PD, presently, there are no cures or promising preventive therapies for PD. This could be attributed to the intricate pathophysiology of PD and the poorly understood molecular mechanism. To address these challenges comprehensively, a thorough disease model is imperative for a nuanced understanding of PD’s underlying pathogenic mechanisms. This review offers a detailed analysis of the current state of knowledge regarding the molecular mechanisms underlying the pathogenesis of PD, with a particular emphasis on the roles played by gene-based factors in the disease’s development and progression. This study includes an extensive discussion of the proteins and mutations of primary genes that are linked to PD, including α-synuclein, *GBA1*, *LRRK2*, *VPS35*, *PINK1*, DJ-1, and Parkin. Further, this review explores plausible mechanisms for DAergic neural loss, non-motor and non-dopaminergic pathologies, and the risk factors associated with PD. The present study will encourage the related research fields to understand better and analyze the current status of the biochemical mechanisms of PD, which might contribute to the design and development of efficacious and safe treatment strategies for PD in future endeavors.

## 1. Introduction

Parkinson’s disease (PD) is the second most prevalent neurodegenerative disorder after Alzheimer’s disease (AD) [1]. It is marked by certain motor dysfunctions involving tremors, rigidity, posture instability, and bradykinesia. Among all neurological disorders, PD is experiencing the most significant growth in terms of deaths, prevalence, and disability-compromised life [2]. A particular gene mutation is frequently the source of familial PD. In contrast, variations in genes that affect PD susceptibility are linked to sporadic PD [3,4].

Even though the symptoms and treatments for PD were first referenced in the “Indian Ayurveda” (5000 BC) and Chinese medical classic “Nei-Jing” (500 BC), James Parkinson, in 1817, was the first to describe them as “the shaking palsy”. According to epidemiological research, PD is a global disease that affects 1–2% of people over 65 and 4–5% of people over 85–89 years old [5,6]. It often strikes people between the ages of 55 and 65. Like AD, 90–95% of cases of PD are sporadic, while 5–10% are familial cases [7]. Familial instances of PD are rare and do not exhibit the typical symptoms of the disease, making it more challenging to comprehend the pathophysiology of PD [3,8]. There is a 1.5-fold higher chance of men than women developing PD [9], and developed countries have reported increased incidences of the disease [10], likely as a result of an increase in the elderly population over there [11].

PD is a condition where the nigrostriatal system that supplies dopaminergic (DAergic) innervation to the striatum has selectively degenerated. A loss of dopamine cells in the midbrain is not the only degenerative process driving PD. The serotonergic, glutamatergic, noradrenergic, and cholinergic systems found in the cortical, brainstem, and basal ganglia regions are among the non-dopaminergic neurotransmitter pathologies in PD [12]. PD also comprises a group of non-motor symptom complexes that may appear years before the motor symptoms [13]. The most prevalent non-motor indications include anxiety, depressive episodes, gastrointestinal issues, sleep difficulties, and olfactory abnormalities [14]. Parkinson’s patients may experience a decline in their quality of life due to non-motor symptoms [15]. Early identification of non-motor symptoms can aid in diagnosing PD and somewhat enhance the patients’ chances of survival [16]. It can take 15 to 20 years or longer for PD symptoms to develop further, though this can vary from person to person [17]. PD patients struggle with verbal proficiency, physical sequencing, transitioning between tasks, and sequenced learning [18]. As illustrated in Figure 1, the primary symptoms experienced by PD patients can be mainly categorized into five types: “early symptoms, primary motor symptoms, secondary motor symptoms, primary non-motor symptoms, and secondary non-motor symptoms” [19]. The neuropathological hallmarks of PD involve DAergic neuronal depletion in substantia nigra pars compacta (SNPC), Lewy bodies (LB) in persisting neurons, and dystrophic Lewy neurites (LN) occurrence [20,21,22]. DAergic neural depletion in the SNPC, leading to a decrease in DA in the striatum, is the primary source of the motor symptoms of PD. The LB formation, intra-cytoplasmic aggregates within the survived neurons, coincides with this reduction in striatal DA levels. Core PD motor impairments are brought on by the dysfunction of basal ganglia circuits caused by depleted striatal DA [23,24].

Although the precise mechanism behind the loss of DAergic neurons in the SNPC is still not completely understood [26], the onset and development of PD may be influenced by misfolding in proteins and their accumulation [27], oxidative stress [28], mitochondrial dysfunction [29], energy deficiency [30,31], excitotoxicity [32,33], cell-autonomous processes [34,35], prion-like propagation of α-synuclein (α-syn) [36,37], protein clearance pathways malfunctioning [38,39], and corticostriatal pathogenesis [40]. Abnormal folding in proteins, followed by aggregation in intracellular portions, has emerged as a principal hypothesis for PD. The LB [41,42] is the main misfolded protein inclusion found in the intracellular spaces of the SNPC in PD. These bodies comprise a variety of misfolded proteins such as ubiquitin (Ub) and α-syn, phosphorylated tau (p-tau), and amyloid-β (Aβ) [43]. The intracellular aggregated α-syn permeates in the cell membrane, leading to neuronal death [44,45]. Despite a noticeable neuronal loss in the SNPC, the central nervous system (CNS) also exhibits widespread neurodegeneration [46,47]. Although the exact molecular mechanisms causing this degenerative disease and its clinical manifestations are still unknown, environmental, genetic, or their combined effects are believed to be responsible.

According to current studies, there may be a connection between commensal gut bacteria and the brain that affects neurodevelopment, brain function, and overall health [48,49]. The microbiome–gut–brain axis is a term used to describe this two-way connection [50]. In addition to the vagal nerve route, immune system mediators, gut-related hormones, and signaling molecules produced from the microbiota could constitute mechanisms that trigger α-syn aggregation via the gut–brain axis [51,52,53]. Enterically generated α-syn’s aggregation and propagation are probably signs of an early pathogenic stage that could subsequently trigger PD motor and non-motor symptoms [54,55]. The genesis of PD may be linked to changes in bacteria that produce short-chain fatty acids and an increase in putative gut pathobionts in the gut microbiome [50]. The gastrointestinal tract’s muscular and secretory functions are controlled by the enteric nervous system (ENS), a nerve network made up of glial cells and neurons [56]. Early ENS changes in Parkinson’s disease have been demonstrated, even before CNS involvement [57,58,59]. Studies conducted on PD patients after death have revealed that some ENS neuronal subtypes aggregate α-syn [60]. Dysregulation of the intestinal microbiota can exacerbate intestinal inflammation, damage to the intestinal epithelial barrier, and the upward diffusion of phosphorylated α-syn from the ENS to the brain. This can also result in a dysregulated gut–brain axis, gastrointestinal dysfunction, and CNS neurodegeneration. PD and intestinal microbiota disorders have a significantly more complicated link than just a one-way causal relationship [61,62,63].

Until it was recognized as the model non-genetic condition, the neuropathology of PD was poorly understood [64]. Though most PD cases are now believed to be idiopathic, 10–20% of patients suffer from Mendelian inheritance-based monogenic PD. Over 90 susceptibility genes have been discovered thus far [2]. The multiple pathways that result in the loss of DAergic neuronal cells and the numerous possible targets for disease-modifying therapy are reflected in the pathophysiology of these genetic contributions. Aging is the most important risk factor for PD. As a result, PD cases are sporadic in adults under the age of 40 and become more common in people in their 70s and 80s [8]. A person’s risk of contracting PD is enhanced if they have one or more close relatives with the condition, although the overall risk is still about 2–5% unless there is a known gene mutation in the family [65].

### 1.1. Risk Factors Associated with PD

#### 1.1.1. Aging

The primary risk factor for neurological disorders is aging. Only 5% of all cases of PD are identified before the age of 60, which is referred to as early-onset PD [66]. Aging decreases patients’ ability to self-heal and raises their risk for neurological conditions. In PD, aging is linked to functional limitations and motor neuron dysfunction [67]. Ageing is linked to increasing oxidative stress and decreased mitochondrial functioning at the cellular level in PD [68]. Studies have established that harmed brain regions of PD had decreased glutathione peroxidase, superoxide dismutase, glutathione reductase, and catalase activity [69]. Researchers also believe that mitochondrial malfunction may produce misfolded protein clumps, which ultimately cause PD, as mitochondrial efficiency falls with aging [70,71].

#### 1.1.2. Gender

Based on different studies, men are more likely than women to develop PD [72]. Uncertainty surrounds the causes of the variations between men and women with PD, while one theory points to the female body’s estrogen-protective properties as the cause. Distinct other proposed theories explaining the difference in PD in men and women include that men are more likely than women to experience mild head injuries, be exposed to industrial pollutants, and might have genes on their sex chromosomes that make them more susceptible to developing PD [73].

#### 1.1.3. Ethnicity

PD is more prevalent in white people than in black people or Asian people, contrary to several studies. Black people and Asian people, rather than white people, possess a 50% lower prevalence of PD. However, the highest prevalence of PD is found among Hispanic people, followed by non-Hispanic white people, Asian people, and black people. One study found that Hispanic people have a higher incidence of PD than black people (10.2/100,000), Asian people (11.3/100,000), and non-Hispanic white people (13.6/100,000), which is 16.6/100,000 people [74,75].

#### 1.1.4. Genetics

A total of 15–25% of patients with PD have a relative who also has the condition. A slightly (2–5%) greater chance of having Parkinson’s exists in individuals with close family members who suffer from the condition. There are several gene mutations connected to PD. While some of these mutations only raise a person’s risk for the disease, others seem to be more causal [76,77]. Early-onset PD is affected by at least five genetic loci and appears to be more frequently associated with specific mutations. The mutations of the *SNCA* protein-coding gene were identified first and have been linked to early-onset PD [78]. *LRRK2* mutations have been connected to late-onset PD and may also be a factor in non-familial PD [79]. The identified mutations included *PINK1* or *PARK6*, DJ-1 (*PARK7*), and Parkin or *PARK2*, all of which have a recessive mode of inheritance [80,81,82]. 90% of instances of PD are sporadic, which means they cannot all be caused by genetics, thus showing that PD has a complex multifactorial etiology [83].

#### 1.1.5. Environmental Factors

The root cause of PD is multifaceted, and it is evident from several studies that certain environmental factors, including exposure to chemicals, like rotenone, paraquat, and trichloroethylene, and lifestyle decisions, like physical activity, alcohol/caffeine consumption, and smoking habit, may contribute to the likelihood of developing the disease [84,85]. Due to their capacity to detrimentally impact brain function, environmental factors such as pesticides, heavy metals, herbicides [86], and head injuries have been implicated in the pathogenetic development of Parkinson’s disease (PD). These environmental elements are recognized for their potential contribution to both the onset and progression of PD, underscoring the intricate relationship between environmental exposures and the development of this neurodegenerative disorder [87,88]. Epidemiological findings indicate that higher concentrations of some metals like mercury in the brain might be related to the onset and progression of PD, even though a direct link between these metals and the disease has not yet been conclusively established. It has been determined that prolonged exposure to harmful heavy metals like lead, mercury, cadmium, manganese [89], and arsenic results in neurotoxicity, which then leads to neurodegeneration [90,91,92,93].

## 2. Genetic Basis of PD

Despite the fact that cases of PD are typically sporadic, various mutations are linked to the disease pathogenesis. They represent roughly 2–3% of late-onset forms of PD and more than 50% of patients with early onset. A clearer understanding of the molecular etiology of hereditary PD has been made possible by discovering gene mutations connected to the onset of familial cases of PD. A long list of genes is known to contribute to PD, and many more may yet be discovered. Mainly, six genes have been explored and linked to heritable PD. Whilst Parkin, DJ-1, *PINK1*, and ATP13A2 mutations are passed down in an autonomously recessive manner, *SNCA* and Leucine-rich repeat kinase 2 (*LRRK2*) mutations are passed down in an autosomal dominant form [3]. General gene-based molecular mechanisms in the pathogenesis of PD have been illustrated in Figure 2.

### 2.1. α-Synuclein

Gene multiplication and point mutation in the *SNCA* gene results in the autosomal dominant PD [104]. However, sporadic PD is believed to occur due to polymorphisms in the *SNCA* gene locus [105]. The *SNCA*/*PARK1* gene encoding α-syn (A53T) was first recognized to cause a familial type of PD because of a missense mutation [106]. Various missense point mutations, namely, A53E, H50Q, A30P, G51D, and E46K, were also discovered shortly after that in the N-terminal region of α-syn. PD linked with *SNCA* frequently develops early and progresses quickly [107,108].

α-syn is a 140-aa protein that has three different domains: a strongly negatively charged C-terminal site, a hydrophobic non-amyloid domain that is capable of adopting a β-sheet conformity [109], and an N-terminus that takes on an α-helical secondary configuration over membrane binding. The protein structure and the positive and negative charges of α-syn are directly associated with pathological modifications of α-syn. The primary area for phosphorylation alterations is the negatively charged carboxyl terminus. By virtue of its hydrophobicity, the core hydrophobic region can easily form a β-pleated sheet. The positively charged amino terminus is vulnerable to acetylation and ubiquitination changes [110,111]. α-syn structural domains are depicted in Figure 3.

The CNS possesses high levels of α-syn expression, which is restricted to the area around synaptic vesicles that might be involved in synaptic transmission [114]. A study on the rat model of PD has shown that the α-syn variants that form oligomers tend to be more cytotoxic than those that form fibrils, resulting in an increasingly serious degeneration of DAergic neurons in the SNPC [115]. LB mainly consists of α-syn, which is phosphorylated at Ser129 of α-syn [116,117]. Amid pathological circumstances, α-syn may adopt a β-pleated secondary framework as the building block for LB and LN [118]. α-syn monomers assemble to generate protofibrils that resemble strings and may expand into bigger fibrils, further worsening the progression of PD pathogenesis [119]. According to certain research findings, oligomeric α-syn species that are toxic to neurons are compartmentalized by LB, suggesting that the aggregates of the protein may not pertain to neuronal toxicity [120,121]. The unfolded protein response (UPR) is provoked by *SNCA* gene triplication in a PD patient [122]. Matsui et al. [123] reported that T64 phosphorylation alters the characteristics of α-syn and promotes the generation of distinctive oligomers in the human PD brain. Such phosphomimetic mutation leads to lysosomal disorder, mitochondrial failure, neurodegeneration, and apoptosis, suggesting the pathogenic potential of α- syn phosphorylation at T64 in PD.

Protein post-translational modifications (PTMs) control protein activity and proteome modifications [124,125]. By changing the α-syn configuration, aggregating kinetics, subcellular localization, fibril ultrastructure, and molecular interactions, these PTMs considerably impact the emergence and dissemination of disease [126]. The combinational implications of PTMs, alongside related non-covalent cofactors on protein functioning, fibril organization, and pathological features, demand additional research, considering α-syn is susceptible to multiple alterations in both PD and non-PD individuals [127]. α-syn binds to the orexin 1 receptor (OX1R), which facilitates the post-translational protein degradation of OX1R through lysosomal and proteasomal pathways [128]. Extracellular signal-regulated kinase and protein kinase B signaling pathways are further downregulated, resulting in orexin neuron destruction that induces sleep behavior disorder, a potential early sign of PD [128]. The spectrin–ankyrin complex, crucial for the precise positioning and functionality of integral membrane proteins like Na1/K1 ATPase, is altered by the binding of α-syn to β-spectrin. Thus, neuronal dysfunction and mortality are caused by a higher level of α-syn in PD and associated α-synucleinopathies [129].

Genome-wide association studies revealed a link between PD and genetic variation in the gene for the tau protein, which is interlinked to AD [130] and regulates cytoskeletal integrity [131]. Interaction between α-syn and tau allows them to mutually promote each other’s aggregation [132]. In an α-syn overexpression model, hyperphosphorylated tau has also been identified [133]. In Drosophila, coexpression of α-syn increased the death of dopamine (DA) neurons induced by tau [134]. α-syn also seems to be involved in mitochondrial malfunction [135]. The chemical reaction of oligomeric α-syn with the membranes of mitochondria may result in their fragmentation and Dynamin-like protein 1 (DLP1)-independent mitochondrial fission [136]. Additionally, α-syn inhibits the functioning of mitochondrial complex I [137].

α-syn mutations potentially harm DAergic neurons, since they change several intracellular signal pathways. The mutations in α-syn A53T have the capacity to block autophagy in transgenic mice’s brains at an early stage and cause synucleinopathy [138]. Furthermore, in mouse models of PD, A53T causes apoptotic pathways in adrenal phaeochromocytoma (PC12) cells that are driven by endoplasmic reticulum (ER) stress and mitochondrial malfunction [139]. α-syn mutant overexpression in DA cells like PC12 and SH-SY5Y severely impairs proteasomal protein cleavage [140]. According to Matsumoto et al.’s [141] studies on CD-1 mice, erythrocyte-acquired extracellular vesicles with α-syn can pass the blood–brain barrier (BBB); this could be an entirely novel mechanism for the brain and peripheral nervous system to communicate during the onset and progression of PD. DAergic neurons die as a result of α-syn selective binding to tropomyosin receptor kinase B (TrkB) and the inhibition of the TrkB signaling pathway in the mouse model studies of PD [142,143].

Moreover, in mouse models of PD, A30P mutation could accelerate the degeneration of DAergic neurons by triggering microglia and increasing the phagocytic oxidase and macrophage-1 expression [139]. Decreasing the expression of α-syn may offer a viable treatment strategy, considering that both mRNA and protein levels increase twofold in mice models of PD [144,145]. Zharikov et al.’s [146] investigation of the rat model of PD stated that disruption of α-syn employing shRNA reduces the progression of motor impairments as well as DAergic neuron degradation. Further, the preclinical studies need to be validated by clinical investigations.

### 2.2. LRRK2

A total of 1–5% of sporadic PD and 5–13% of familial PD is associated with mutations in *LRRK2* [147,148]. The *LRRK2*/*PARK8* gene mutation, which results in autosomal dominant PD, carries the highest risk of familial PD [17,149]. I2020T, R1441G, G2385R, R1441C, R1628P, R1441H, G2019S, and Y1699C are seven of the documented missense *LRRK2* variants that have been confirmed to be pathogenic. These mutations are found in various functional domains of *LRRK2* [150,151]. It is fascinating to note that variations in *LRRK2* seem to be population-specific [152,153,154,155,156,157,158,159,160]. *LRRK2* is large in size with 2527 amino acids and is made up of multiple functional domains [161]. *LRRK2* has two enzyme-like functions as its catalytic center, which comprises the Ras of complex (ROC) and the C-terminal of the ROC bidomain with the kinase region [162,163]. The structural domain representation of *LRRK2* is given in Figure 4. *LRRK2* is abundantly expressed in organs like the kidney, lungs, heart, and brain [164]. Additionally, it has been observed that *LRRK2* can be identified in monocytes, lymphocytes, blood, cerebrospinal fluid (CSF), and urine [165]. Lipid dynamics are essential for vesicle trafficking, lipid metabolism, and lipid storage, all of which depend on *LRRK2* substrates from the Ras-associated binding (Rab) GTPase family. Furthermore, *LRRK2* is also linked to the phosphorylation and activity of enzymes that catabolize lysosomal lipids and the plasma membrane [166]. Elderly *LRRK2* G2019S mutant carriers have substantially higher rates of PD morbidity [167].

In DAergic neurons and cultures of primary neuronal cells of a PD brain, *LRRK2* G2019S mutation increases α-syn mobility and boosts α-syn accumulation [170]. The mutation has also been found to contribute to tau protein neural pathology in *LRRK2*-linked PD by promoting tau transmission in neuronal cells in mice [171]. *LRRK2* interferes with DA signaling in addition to limiting neuronal survival. Through phosphorylation of apoptotic signal-regulating kinase 1 at the Thr832 site and boosting the kinase potential, *LRRK2* performs significant roles in neuronal death [172]. *LRRK2* has a variety of functions in the secretory pathway and might assist in DA signaling in a mice model of PD [173]. *LRRK2* may cause neurons to die by suppressing myocyte-specific promoter factor 2D action, which is necessary for neural cell survival [174]. Dopamine receptor D1 uptake is compromised by the *LRRK2* G2019S mutation, which alters signal transduction [175]. Additionally, the G2019S mutation increases kinase activity, impairing the synaptic vesicle transportation in ventral midbrain DAergic neurons [176]. A close correlation exists between mitochondrial dysfunction and the *LRRK2* G2019S mutation. According to Howlett et al. [177], this mutation causes mitochondrial DNA (mt-DNA) destruction, which is reversible through a pharmacological decrease in the activity of *LRRK2* kinase. Triggering mitochondrial DLP1 levels and neural toxicity, mitochondrial disintegration and dysfunction are caused by *LRRK2* G2019S’s ability to bind to and strengthen its interaction with mitochondrial DLP1 [178,179].

### 2.3. PINK1

A 581-aa serine/threonine kinase is encoded by *PINK1* [180]. The regions at the N-terminal oversee *PINK1* processing and transport to mitochondria. The kinase domain consists of two lobes, N and C. A significant number of PD-linked mutations and well-characterized phosphorylation sites are in this *PINK1* domain. *PINK1* structural organization is depicted in Figure 5. The autosomal recessive *PARK6* form of PD is brought on by *PINK1* deficit [181]. Inner mitochondrial membrane-bound proteases cleave *PINK1* at A103 and F104 to a 52-kD split, which escapes into the cytoplasm and is deteriorated by ubiquitination [182]. As a result, *PINK1* basal levels are not detectable. Transport via the outer membrane of mitochondria is hampered by mitochondrial stressors like depolarization of the membrane, electron transport chain (ETC) unit malfunction, and mutagenesis stress, which stops proteolysis. This causes *PINK1* buildup on the outer membrane of mitochondria, which triggers dimerization and activates the kinase domain [183]. As a result, *PINK1* serves as a sensor for mitochondrial damage by turning on pathways for mitochondrial quality monitoring. Disruption of *PINK1* and Parkin also causes an imbalance in the activity of the inositol 1,4,5-triphosphate receptor, which sharply increases calcium release from the ER [184].

By phosphorylating LETM1 at Thr192 to promote Ca^2+^ release while facilitating its transportation, *PINK1* depletion is connected to mitochondrial failure and mitochondrial Ca^2+^ dysregulation [187]. According to Martinez et al. [188], the deregulation of the misfolded protein response of mitochondria relying on the *PINK1* homolog causes non-apoptotic degeneration of DAergic neurons. A dominant-negative pathway may raise the risk of PD in heterozygous *PINK1* G411S mutation carriers, as this mutation drastically reduces *PINK1* kinase function [189]. In PD mice, *PINK1* mutations promote the buildup of defective mitochondria with aging, stimulating the misfolded protein response of mitochondria and prolonging life [190]. The *PINK1* mutations I368N and Q456X decrease either protein stability, levels, or kinase activity, raising the likelihood of PD [191]. Parkin is less likely to be recruited to depolarized mitochondria by other mutations, such as G309D, A168P, L347P, and H271Q [192].

### 2.4. Parkin

Originally thought to function as ubiquitin E3 ligase that could be stimulated by autophosphorylated *PINK1* [193], Parkin is a protein that is encoded by *PARK2* and is necessary for the degradation of target molecules through the ubiquitin–proteasome system [194]. The structural domains of Parkin are shown in Figure 5. A total of 10–25% of early-onset PD occurs due to Parkin gene mutations [195]. In order to facilitate the mitophagy destruction of defective mitochondria, *PINK1* aggregates on the malfunctioned mitochondrial membrane stimulate Parkin E3 ubiquitin ligase operation and recruit cytoplasmic Parkin molecules to the dysfunctional mitochondria [196]. By encouraging Parkin recruitment to mitochondria, reactive oxygen species (ROS) also trigger *PINK1*/Parkin pathway-controlled mitophagy [197]. A defect in Parkin recruitment for depolarizing mitochondria results from mutations such as C441R, R42P, C289G, R46P, C253Y, C212Y, and K211N, which substantially restrict mitophagy [198]. Mitophagy and mitochondrial biogenesis are coordinated by the Parkin interacting substrate (PARIS) axis. A network of sound mitochondria is maintained by basal state cellular homeostasis. A transcriptional program involved in mitochondrial biogenesis is connected to mitophagy. Parkin and PARIS are involved in one pathway in this intricately regulated process. The steady-state levels of PARIS are regulated by Parkin-mediated ubiquitination, which is aided by *PINK1* as a priming kinase. Parkin expression and activity are increased by mitophagy triggers, which causes PARIS to be broken down by proteases. Reduced levels of PARIS alleviate Peroxisome proliferator-activated receptor-gamma coactivator-1 alpha (PGC-1α) transcriptional inhibition, hence facilitating mitochondrial biogenesis [199].

### 2.5. DJ-1

Early onset of recessive PD has been linked to mutations in DJ-1, which is encoded by the *PARK7* gene [200]. A PD-linked gene retained in both prokaryotes and eukaryotes, DJ-1 (*PARK7*), is an evolutionary ancient gene [96,99,201]. The functional domains of DJ-1 are shown in Figure 6. The function-related mutational loss that affects DJ-1 protein integrity and homo-dimerization causes *PARK7* PD, which is inheritable in an autosomal recessive manner [202]. Several DJ-1 mutant variants have been linked to PD, including L166P, M26I, L10P, and P158. It has been reported that a DJ-1-associated PD brain has neuropathological issues as per the genetic data, indicating a distinctive mutation of L172Q in the *PARK7* gene [203]. In cells overexpressing *PINK1*, DJ-1 might interact with and stabilize *PINK1* [204]. DJ-1 also engages with α-syn, which modifies its accumulation by establishing a weak hydrophobic interaction [205] and reversing α-syn-mediated toxicity to cells [206]. Along with *PARP1*, DJ-1 preserves genomic stability, and disruption of this connection might have an influence on DNA damage accumulation, impaired DNA repair, and, ultimately, neurodegeneration. Therefore, faulty DNA repair is associated with the PD pathophysiology brought on by DJ-1 mutations [207].

The antioxidant capabilities, antiapoptotic implications, and impact on mitochondrial respiration, shape, shifts, and biosynthesis are a brief overview of the neuroprotective actions exerted by DJ-1 [209]. DJ-1 mutations principally affect a protein that facilitates intracellular oxidation–reduction [210]. Due to its elimination of neurological protective effects against H_2_O_2_ and elevation of thioredoxin-1 by suppressing the nuclear factor erythroid 2-related factor 2 signal pathway, mutant DJ-1 (L166P and M26I) raises the vulnerability of SH-SY5Y cells to oxidative stress [211]. Additionally, the D149A mutation ends this shielding [212]. L172Q, L10P, and P158∆ are the three mutations that contribute to a decrease in the stabilization of proteins [213]. In vivo experiments have demonstrated that recombinant DJ-1 can also stop the DA degeneration caused by 6-hydroxydopamine (6-OHDA) or α-syn [214].

### 2.6. Vacuolar Protein Sorting 35 (VPS35)

The endolysosomal network is made up of many tubulovesicular organelles that are important for protein manufacturing, nutrition intake, cellular trafficking, and apoptosis [215,216]. The heterotrimeric complex, comprising *VPS26*, *VPS29*, and *VPS35*, is a crucial component of the endolysosomal system’s sorting process [217,218,219]. *VPS35* structural domains are represented in Figure 7. The mechanism of transmembrane protein shifting between Golgi and endosomes is regulated by *VPS35* [220,221]. The *VPS35* gene was originally linked to late-onset PD in an Austrian family. It contributes 1% to familial PD. The *VPS35* D620N mutation is harmful in PD patients from American, European, and Asian families [222]. In PD-afflicted fibroblasts, the *VPS35* D620N mutation decreases enzyme functionality in complex I and II, resulting in mitochondrial failure by reusing DLP1 complexes [136,223]. Finally, DAergic neuron loss occurs as a result of mitochondrial malfunction brought on by *VPS35* deficiency [224]. PD-related *VPS35* mutation, R524W, hinders retromer endosomal interaction and causes α-syn to aggregate [225]. In a Drosophila model, the *VPS35* P316S mutation also caused some PD symptoms, such as decreased climbing power and a reduction in Daergic neurons, which made the fly particularly susceptible to the drug rotenone [226].

### 2.7. Glucocerebrosidase 1 (GBA1)

The 497-aa protein β-glucocerebrosidase (GCase1) is encoded by *GBA1*, which is located on the lysosomal membrane. *GBA1* structural domain organization is illustrated in Figure 8. The most prevalent gene-associated PD risk factor is the catabolism of the glycolipid glucocerebroside into glucose and ceramide in the lysosome [227,228], which causes Gaucher disease. In 7–12% of patients, heterozygous *GBA1* mutations have been identified as the most prevalent genetic risk factors for PD [229]. As patients with PD or *GBA1* mutations may display extensive LB or LN, GCase1 malfunction and α-syn pathogenesis appear to be related inextricably [121]. In addition, GCase1 enzyme activity is decreased in *GBA1*-PD and sporadic PD cases [230,231] and has an inverse relationship with the degree of α-syn pathology [232].

In the brain and CSF of PD patients, an enormous decline in GCase 1 functioning and protein concentrations has also been identified [234,235]. Employing a small-molecule modulator to activate GCase 1 reinstated lysosome functionality and eliminated the buildup of pathogenic α-syn in PD individuals, suggesting the plausible role of GCase 1 in the onset of idiopathic PD [236]. According to estimates, L444P and N370S are the two most prevalent *GBA1* mutations, accounting for around 10% to 25% of PD instances [237,238]. The instability of α-syn tetramers is reversed, and human DAergic neurons are protected against α-syn prefabricated fibril-initiated toxicity by inhibiting glycosphingolipids accumulation and overly expressed *GBA1* to increase GCase 1 function [239]. The mutation of N370S caused GCase 1 to be retained in ER, stopping its flow to the lysosome, which activated the UPR and caused the Golgi apparatus to fragment by ER stress activation [240,241]. In addition, heterozygous L444P *GBA1*-mutated murine neurons showed dropped α-syn tetramers along with associated multimers [239], and L444P *GBA1* mutation provoked α-syn-mediated DAergic neuronal loss in the SNPC of mice models of PD [242].

## 3. Cellular and Molecular Mechanism Underlying the Pathogenesis of Neurodegeneration in PD

### 3.1. Protein Misfolding and Aggregation

The aggregation of protein is a biological process characterized by the buildup of misfolded proteins inside or outside of cells [243]. These protein aggregates are the pathogenic hallmark of PD, indulging the LB-like α-syn protein buildup [244].

In the investigation of PD using mice models conducted by Jones et al. [245], the pathogenic misfolding, ensuing accumulation, and buildup of α-syn are essential components of the disease pathogenesis. Notably, serotonin plays a crucial role in significantly facilitating the regulation of adult hippocampal neurogenesis [246]. In the rat model of PD, α-syn detrimentally impacts hippocampal neurogenesis. This effect is accompanied by diminished 5-HT neurotransmission, occurring prior to the onset of aggregation pathophysiology and motor impairments [247]. Oxidative stress triggers the non-receptor tyrosine kinase c-Abl, whose crucial function is proposed in α-syn-initiated neurodegeneration [248]. α-syn phosphorylation at tyrosine 39 occurs due to c-Abl activation, potentially contributing to the disease development in hA53Tα-syn transgenic mice. Additionally, it has been established that c-Abl inhibition might deliver protection against neuronal degeneration induced by α-syn [249]. A number of other kinases, including G-protein-coupled receptor kinases [250], polo-like kinases [251], casein kinase II [252], and *LRRK2* [253], also phosphorylate α-synuclein Ser129. α-syn protein misfolding in the pathogenesis of PD is represented in Figure 9.

*PINK1* becomes stable on the membrane of mitochondria from the outer side and activates Parkin ubiquitin ligase functionality through ubiquitin phosphorylation at Ser65 during mitophagy, which is triggered by the misfolded protein aggregation or the mitochondrial membrane potential loss [254]. It causes ubiquitin chains to assemble on the outer mitochondrial membrane, resulting in autophagy receptor activation in the process. Pathogens are eliminated through autophagy by engulfing the intracellular pathogens in autophagosomes and transferring them to lysosomes for destruction. Even though autophagy receptors p62 and optineurin are known for binding ubiquitin chains on dysfunctional mitochondria, their precise function in mediating mitophagy is still unknown [255].

DJ-1 suppresses α-syn aggregation and guards neurons against intracellular oxidative circumstances [256,257]. The main cause of α-syn aggregation in pathological circumstances is Ser 129 phosphorylation [258,259,260,261]. Although the PD pathogenesis that is *LRRK2*-linked is not completely known, the pathogenesis of PD has been found to contain *LRRK2* mutations that result in the generation of aggregated protein and the degeneration of the neurons. The clearance mechanisms of UPS and autophagy carry out the protein buildup management. An in vivo study with zebrafish indicated that the overexpression of *LRRK2*, along with its interaction with the UPS, results in the accumulation of proteins [262]. It has been observed that amplified expression of *LRRK2* hinders the production of aggresomes promoted by MG132, which is necessary for autophagic breakdown. In differentiated SH-SY5Y cells, its dysfunction also causes an aberrant buildup of protein aggregates and worsens the cytotoxicity caused by proteinopathy [263]. In the study by Liu et al. [264], it is proposed that synphilin-1 may possess a neuroprotective impact by reducing the PD-like phenotypes caused by mutant *LRRK2*. The excessive accumulation of proteins, triggered by a malfunctioning UPS and a compromised cellular physiological system, may collectively contribute to the pathophysiology of PD in Drosophila. Since the ER regulates the folding and sorting of protein, such protein buildup could potentially be caused by dysfunction of this organelle.

### 3.2. ER Stress

The observed death of DAergic neurons in the SNPC of PD patients is also associated with ER stress [265]. According to different experimental data, the misfolded/ unfolded protein emergence causes ER stress, which aids in the death of neurons or apoptosis [266,267,268]. The ER-related UPR is stimulated because of misfolded protein aggregation. The initial goal of the UPR is to normalize the cell state by interfering with protein translation, eliminating misfolded proteins, and activating signaling pathways that create molecular chaperones accountable for the folding of proteins. However, under prolonged disruption, the UPR shifts its focus to cellular apoptosis, as observed in the rat model of PD, which further needs to be validated by clinical studies [269]. UPR is a complicated response that employs a variety of mechanisms to lessen the load of abnormal proteins [270]. Three transmembrane proteins, including the activating transcription factor 6 (ATF6), PRKR-like ER kinase (PERK), and inositol-requiring enzyme 1 (IRE1), are activated in the central nervous system to begin the UPR process [271]. When a cell is stressed, these transmembrane proteins separate from glucose-regulated protein 78 (GRP78) and trigger several intracellular signals that reduce the load on the ER. These intracellular signals include transcriptional and translational inhibition which decrease the ER concentration of proteins and boost the number of molecular chaperones to improve the ER capacity for folding [272]. The peripheral inflammation in PD is exacerbated by lymph node swelling, which is directly associated with macrophage activation and is brought on by meningeal lymphatics discharging oligomeric α-syn [273,274]. ER stress caused by oligomeric α-syn is amongst the most plausible causes of PD. Pathologic molecular mechanisms of ER stress in PD are depicted in Figure 10.

It is believed that mild ER stress plays a neuroprotective role. Nevertheless, neurons typically die or undergo apoptosis in response to severe ER stress. The most conserved UPS and a crucial UPS regulator is IRE1/ X-box binding protein 1 (XBP1) [122]. The DAergic neuronal death brought on by 6-OHDA injections is prevented by the conditional deletion of XBP1 in a mice model of PD [275]. ER stress signaling is increased in PD models containing Parkin and *PINK1* mutations, while the suppression of PERK is neuroprotective [276]. Since the expression of the chaperone hsp70 may mitigate neural mortality in rat and mouse models of PD, ER stress-related enhanced chaperone expression seems to be advantageous for cells [277,278]. Human α-syn overexpression induces ER stress, upregulating proapoptotic C/EBP homologous protein (CHOP) in DAergic neural cells in the SNPC by triggering the PERK and ATF6 signaling pathways. This is inhibited by GRP78 [279,280]. The *GBA1* gene N370S mutation results in a considerable decrease in enzyme functioning and GCase 1 protein, as well as retention inside the ER, which obstructs its movement to the lysosome; following this, ER stress is activated, with triggered UPR and disorganized Golgi apparatus [241]. A recent study using intraneural Tunicamycin injection as a new model of PD demonstrated that ER stress might serve as an essential part in the development of PD by replicating some of the phenotypic traits seen in rat models of PD [281]. Therefore, more research is required to ascertain how ER stress and the UPR influence the survival and death of neurons.

### 3.3. Calcium Homeostasis

Different voltage-gated calcium channels are believed to play a role in calcium homeostasis control. These channels and pumps are placed on the plasma membrane in order to transfer the calcium both within and outside the cell [282]. They also ensure the close functional connection for cellular physiology, and any shift in this regulation results in impaired calcium homeostasis, which might then trigger the death signaling pathways [283]. Being the primary calcium reservoir, the ER controls folding proteins and keeps calcium homeostasis stable. In PD, there is a disruption in the interaction between mitochondrial calcium signaling and the ER, which results in neuronal death [284,285]. α-syn mutations have been identified to cause calcium overstimulation and cell death [286,287]. Ca^2+^ could attach itself to the C-terminus of α-syn, control its secretion, and encourage the assembly of α-syn aggregates. α-syn has a role in synaptic vesicle endocytosis and is primarily found at presynaptic terminals, which are sites of significant Ca^2+^ fluctuations. The N- and C-terminus of α-syn engage in interactions with isolated synaptic vesicles, but Ca^2+^ regulates the binding with the C-terminus, enhancing α-syn’s ability to bind with lipids [288]. It has been proposed that α-syn and Ca^2+^ influence vesicle pool homeostasis via two different mechanisms in a mouse model of PD: first, by encouraging intervesicle interactions; second, by binding synaptic vesicles to the plasma membrane that potentially modify their ability to access voltage-gated Ca^2+^ channels [289]. In PD, nigral DAergic neurons die because of mitochondrial Ca^2+^ signaling, primarily through regulation of ATP synthesis and mitochondrial oxidative stress. It has been discovered that in the SNPC neurons, a rise in cytosolic Ca^2+^ is dependent on α-syn, which is accompanied by an increase in mitochondrial Ca^2+^ and mitochondrial oxidation [290]. It has been observed that a cytosolic Ca^2+^ rise brought on by L-type Ca^2+^ channel-mediated Ca2^+^ influx stimulates mitochondrial ATP production in cultured hippocampus neurons [291]. In PD patients with the N370S mutation in β-glucocerebrosidase, lysosomal Ca^2+^ store concentration have also been reported to be reduced, accompanied by changes in lysosomal form [292].

### 3.4. Dopamine Metabolism and Toxicity

PD has been demonstrated to cause a substantial decline in the DA-producing cells in the SNPC that maintains Parkinson’s tremor, rigidity, impaired balance, slowed movement, and coordination [293,294]. DAergic neurons may suffer oxidative damage brought on by DA oxidation, as shown in Figure 11. Mitochondrial oxidative stress results in the buildup of oxidized DA, leading to lowered glucocerebrosidase activity, dysfunctional lysosome, and the α-syn buildup in PD neurons [295]. Overexpression of DA transporters, leading to increased DA re-uptake and elevated cytosolic DA levels, induces the degeneration of dopaminergic (DAergic) neurons in mouse models of PD [296,297]. The cytoplasmic formation of aminochrome during the DA auto-oxidation of neuromelanin may cause DAergic neuronal toxicity [298,299]. Additionally, DA stimulates neurodegeneration in the SNPC of mice models of PD and drives the synthesis of soluble A53T α-syn oligomers [300]. Moreover, this needs to be validated through clinical studies. In the mitochondrial extracellular membrane, monoamine oxidase catalyzes the transformation of cytosolic DA to a PD-linked endogenous neurotoxin [301]. This process results in a lengthy accumulation of glyceraldehyde-3-phosphate dehydrogenase and causes irreversible inhibition of the enzyme activity, which harms neurons due to ROS production [302]. In neurons with a *PINK1* defect in the mouse model of PD, DA caused mitochondrial permeability transition pore opening by producing ROS [303]. DA buildup from extracellular DA reuptake into the cytoplasm causes DA neurotoxicity. DA neurotoxicity can be prevented by interfering with the chemical relationship between the dopamine D2 receptor and DA transporter [304]. These investigational studies and facts suggest that chemical intermediates formed from DA are one of the primary causes of PD, which further needs to be validated by clinical studies.

### 3.5. Mitochondrial Dysfunction

Nearly all neurodegenerative conditions, including PD, have been linked to mitochondrial abnormalities [305,306]. It is connected to the drop in adenosine triphosphate levels. Since it produces ATP, a source of chemical energy for cells, the mitochondria are considered a cell powerhouse. ROS are produced in cells with defective physiology because impaired mitochondrial function lowers the quantity of ATP. Due to their high metabolic activity and reliance on aerobic metabolism, the physiology of neurons is severely affected by any mitochondrial dysfunctioning [307,308,309]. The pathogenesis of PD occurring due to dysfunctional mitochondria and its related molecular pathways is depicted and explained in Figure 12. Norberg et al. [310] have shown that mitochondria are also essential for controlling apoptosis. Mutations associated with PD may lead to mitochondrial dysfunction. Prefibrillar α-syn oligomers that were soluble displayed a number of the mitochondrial dysfunctional characteristics seen in PD cell models, including increased cytochrome c release, altered potential of the membrane, dysfunction of mitochondrial complex I, and disrupted Ca^2+^ homeostasis [311]. A lipid peroxidation byproduct called 4-hydroxynonenal encourages intracellular buildup, extracellular vesicle ejection with toxic α-syn, and subsequent incorporation into nearby neurons, which leads to PD development in rat and mouse models [312]. Recently, some in vitro and animal studies have reported that α-syn-induced mitochondrial ROS generation is enhanced by high-temperature requirement serine protease A2 (HtrA2) knockdown, which activates microglial cells, suggesting the possible role of HtrA2 in PD pathogenesis [313,314]. However, clinical validation needs to be established to confirm the role of HtrA2 in PD.

In vitro and in vivo mitochondrial fragmentation and neural death are caused by PD-associated *VPS35* gene mutation, which encodes a crucial retromer complex component owing to enhanced binding to DLP1 [136]. The mt-DNA transcription factor expression is declined in the SNPC of patients with idiopathic PD. Real-time PCR study shows that PD patients have fewer transcription/replication-related molecules and fewer replicas of mt-DNA [315]. In SH-SY5Y, silencing the *GBA1* gene or inhibiting the GCase 1 enzyme functioning leads to mitochondrial malfunction as well, resulting in decreased respiratory chain activity in the mitochondria, in addition to mitochondrial depolarization and fragmentation linked to elevated ROS levels [316]. PD neurotoxic MPTP is converted into its active metabolite MPP+ which is then transported into DAergic neurons [317]. MPP+ is regarded as a mitochondrial complex I inhibitor that limits ATP synthesis and induces the formation of oxides and nitrites, ultimately destroying DAergic neurons in the mice model of PD [318,319]. Mice become more susceptible to MPTP when DAergic neurons have mitochondrial complex I partial deficiency [320]. Potential interactions exist between genetic risk factors and mitochondrial dysfunction brought on by environmental neurotoxins in PD etiology. The downregulation of DJ-1 may cause the susceptibility of DAergic neurons brought on by the injection of subtoxic MPTP, which also increases the toxic effects of mutant α-syn [321]. A long non-coding RNA stimulates the expression of *LRRK2*, which aids in promoting MPTP-induced parkinsonism [322].

### 3.6. Oxidative Stress

The electron transport chain (ETC) within mitochondria is a primary source of ROS, making it a pivotal target for the adverse effects of ROS. Mitochondria possess a defense mechanism to neutralize ROS and repair ROS-induced damage, acting as a protective barrier against oxidative harm to cells. ROS formation and the subsequent release of proapoptotic proteins from the intermembrane space of mitochondria can activate distinct apoptotic pathways, as illustrated in Figure 13 [323]. It is strongly proposed that the formation of ROS causes oxidative stress, which is a factor in the neurodegenerative processes [324,325]. The primary site for ROS generation is mitochondrial complex I [326,327]. ROS generation explicitly damages complex I, leading to its inhibition and further amplifying ROS production [325]. Under normal circumstances, defense mechanisms work to mitigate the negative impacts of ROS. However, when the balance between ROS generation and antioxidant defense is disrupted, excessive ROS production results in oxidative damage [156].

Excessive ROS can inflict damage on all macromolecules, including lipids, proteins, and nucleic acids, as well as enzymes, leading to a significant decline in physiological activity. A metabolomic study by Lan et al. [328] revealed a link between Parkinson’s disease (PD) and dyslipidemia, suggesting that the stimulation of sphingolipid metabolic pathways may also contribute to PD etiology. Jacquemyn et al. [329] emphasized the substantial role of lipids in PD genesis, highlighting connections between the functions of two genes associated with the disease through neural lipid metabolism. In the CNS, DAergic neurons are particularly susceptible to oxidative stress, ultimately culminating in cell death and the pathophysiology of PD [157]. Changes in calcium homeostasis and inflammatory responses are two additional pathways that may be influenced by oxidative stress [330]. Notably, one of the elements triggering redox conditions in specific brain regions may be DA itself. Experimental lesioning of DAergic neurons using one of the DA derivatives, 6-hydroxydopamine [331], indicates that while synaptic vesicles efficiently sequester DA, cytosolic DA can inflict damage on neurons [332]. Consequently, a wealth of evidence underscores the significant impact of oxidative stress on the pathophysiology of PD.

### 3.7. Nitrosative Stress

Nitric oxide (NO) is one of the reactive nitrogen species (RNS) that have been implicated in degeneration pathways [333]; by encouraging α-syn misfolding and boosting the buildup of α-syn aggregates, RNS increases the rate at which the disease progresses. The sheer volume of s-nitrosylated proteins accumulating in LB clusters indicates that once activated, RNS is speeding up the aggregation of several proteins essential for the ongoing existence and functioning of neurons in addition to α-syn [309]. Research on postmortem studies in PD patients has revealed signs of significant nitrosative stress in addition to oxidative damage [334]. Tyrosine residues on α-syn that have been nitrated result in increased aggregation and decreased proteasomal breakdown [335]. S-nitrosylated Parkin is more prevalent in the brain of PD patients, and this version of Parkin has lower enzymatic performance and fewer neuroprotective properties [336]. Elevated amounts of s-nitrosylated protein disulfide isomerase (PDI), which plays a role in protein misfolding and neurodegeneration, have been found in susceptible neurons in PD brains. PDI loses some of its neuroprotective properties when it is s-nitrosylated. Additionally, NO and superoxide can combine to generate peroxynitrite, which possesses higher cellular toxicity potential and worsens the consequences of oxidation [337].

### 3.8. Apoptosis

Apoptosis is one of the primary causes of neural cell death in PD [325,338,339]. Cell shrinking, chromatin condensate, membrane blebbing, and chromosomal and nuclear DNA fragmentation are all signs of apoptosis [340]. Additionally, postmortem and in vitro studies that showed higher expression of active caspase-3 in the SNPC have validated apoptotic significance in the etiology of PD [341,342,343]. Activating caspases is a key step in the progressive sequence of events known as apoptosis [344,345]. Two distinct mechanisms might start caspase activation: the intrinsic pathway, often known as the mitochondrial pathway, and the extrinsic pathway [346,347]. The intrinsic apoptotic pathway is assumed to be the leading cause of neural death [348,349]. The buildup of iron hinders the function of insulin-like growth factor 2 (IGF2) and the transcription factor, zinc finger protein 27 (ZFP27). This, in turn, diminishes autophagy induced by Microtubule-associated protein 1 light chain 3 (LC3), ultimately reducing dopaminergic (DAergic) neurons in mice models of PD. These molecular processes contribute to the progression of PD [350].

It is still unknown how the several pathogenic mechanisms associated with PD, including malfunctioned mitochondria and α-syn accumulation, engage with each other, resulting in apoptotic cell death. In SHSY cells overexpressing A53T, either the wild-type or mutant form of α-syn, and in mitochondria of isolated rat brains, it has been observed that α-syn localizes on the mitochondrial membrane. In in vitro systems, it has been proposed that this interaction causes oxidative stress and cytosolic release of cytochrome c [351]. Cytochrome c causes mitochondrial-aided apoptosis after entering the cytoplasm and interacting with prosurvival, antiapoptotic proteins in vivo [351,352]. The injection of Cu/Zn SOD prevents apoptosis from occurring, while NGF deficiency causes DNA breakage and enhances ROS generation in sympathetic neurons [353]. According to Wang et al. [354], supplementing pretreatment neural stem cells with glial cell line-derived neurotrophic factor (GDNF) at 1 and 6 h showed defense against oxygen–glucose starvation, pointing to the neuroprotective effect of GDNF in neurodegeneration in a rat model of PD.

### 3.9. Neuroinflammation

Neuroinflammation is characterized as an inflammatory response of brain tissue caused by immune cells and the mediators that they release, such as chemokines, cytokines, ROS, and additional secondary transmitters [355,356,357,358]. Neuroinflammation is also caused by environmental factors and genes which significantly impact immune system activation and modification. Microglial activation is linked to the occurrence of α-syn pathological conditions, which can be triggered by inflammation.

Neurotoxicity ensues from the release of detrimental proinflammatory cytokines, notably interleukin-1β (IL-1β) and tumor necrosis factor-α (TNF-α), by activated microglia. This inflammatory response, coupled with the generation of free oxygen radicals, contributes to the harmful effects on neural tissues. The combined actions of these activated microglia-derived factors can lead to cellular damage and dysfunction, exacerbating the neurodegenerative processes associated with various neurological disorders [359,360]. Misfolded α-syn and other inducers, like proinflammatory bacterial byproducts, can activate microglia [361,362]. The antigen presentation to T cells by microglia, which is dependent on major histocompatibility complex type II (MHCII), is another way that T cell-mediated malfunctioning actively contributes to neuronal cell death [363,364]. B cells are crucial for adaptive immunity and may have a part in PD etiology. T helper cells induce B cells to develop into plasma cells, which produce immunoglobulins visible in PD patients’ brain tissue with neurodegeneration [365,366]. Molecular mechanisms of neuroinflammation in PD pathogenesis are displayed in Figure 14.

### 3.10. Immune System Deregulation

Research evidence suggests that the immune system also contributes to the onset of PD. Proinflammatory cytokines like IFN-γ, IL-6, IL-1, and TNF-α are increased in both the CSF and the brain of PD patients postmortem [367,368,369]. Numerous investigations, including those using MPTP, 6-OHDA, and rotenone in the rat model of PD, have shown microglial activation caused by neurotoxins in PD models [370,371,372]. The immune system and α-syn are closely related [373]. Activation of microglia also occurs in the striatum of the rat model of PD because of α-syn overexpression, along with an increase in IL-1β, TNF-α, and IFN-γ (Figure 8) [374]. The absence of IFN-β activity is linked with higher amounts of LB that contain α-syn, a decrease in DAergic neurons, and disruption of DA signaling in the SNPC. On the other hand, in a familial PD mice model, IFN-β overexpression prevents DAergic neurons’ death [375]. Human SNPC neurons exhibit MHCI, and DAergic neurons produced from human stem cells can induce it. To activate T cells and trigger autoimmune reactions, foreign ovalbumin is absorbed by DAergic neurons that produce an antigen derived from it. This causes DAergic neurons to die [376]. Mhc2ta genetic allelic variations control the expression of the MHCII, which regulates the activation of microglia and DAergic neurons caused by α-syn [377]. MHC-II expression, the entry of proinflammatory peripheral CCR2+ monocytes into the SNPC, and the consequent degeneration of DAergic neurons are all facilitated by α-syn [378]. *PINK1* insufficiency functions as an early regulator of neuronal innate immunity, since *PINK1*/Parkin might lower inflammatory circumstances that initiate immune response-eliciting pathways by cutting mitochondrial antigen presentation in the mouse model of PD [379,380]. The pathogenesis of people with sporadic PD forms and *LRRK2* G2019S mutations includes the immune system and endocytosis [381]. The differentiation of immunological homeostasis and bone marrow myelopoiesis may be affected by *LRRK2* G2019S [382]. Finally, we can say that parkinsonism is caused by an immune system that is out of balance or dysregulated and that PD is an autoimmune condition.

### 3.11. Non-Motor Pathologies in PD

Non-motor symptoms such as depression, constipation, sleep difficulties, and hyposmia can sometimes precede many of the motor symptoms of PD over the years. These non-motor characteristics are seen in a wide range of PD patients, including those with hereditary PD etiology [383]. More than 90% of PD patients experience hypo- or anosmia, which is typically bilateral and may appear before dopamine deficiency-related motor symptoms manifest [384]. According to Braak et al. [385], the evolution of Lewy bodies’ dispersion and spread from the lower medulla in PD may be reflected in the emergence of hyposmia and sleep behavior disorder. On the other hand, cholinergic denervation, the advent of cognitive impairments, and dementia may potentially be connected to olfactory dysfunction in the later stages of Parkinson’s disease [386,387].

One of the main categories of PD’s non-motor indications is ocular abnormalities. Although it might be difficult to interpret, blurry vision is frequently linked to PD. Ocular symptoms, including double vision, fuzzy vision, wet eyes, and visual hallucinations, are the most frequently reported ones [388,389,390]. The fact that ophthalmological conditions raise the risk of falls and injuries associated with falls together with postural and gait instability highlights the potential impact [391]. From the prodromal premotor phase to the late stages of the disease, neuropsychiatric traits, including depression and anxiety, are present in PD; these features are correlated with the motor state, with anxiety being more prevalent during off periods. Anxiety and depression frequently coexist, and understanding the anxious depressive phenotype is crucial for efficient treatment [392,393]. In people with PD, constipation may result from the autonomic nervous system not functioning properly, which causes the intestines to function slowly and cause constipation [394]. Understanding the intrinsic and extrinsic mechanisms of constipation linked to PD addresses the pathophysiology of constipation in PD [395].

### 3.12. Non-Dopaminergic Pathologies in PD

The brain’s noradrenergic system distributes the neurotransmitter noradrenalin across the brain via a network of efferent projections. It is essential for cognitive functions and may be involved in both the motor and non-motor symptoms of PD. In PD pathophysiology, a profound loss of noradrenergic circuits has been observed [396,397]. The noradrenergic system also has an anti-inflammatory and neuroprotective impact on dopaminergic degeneration. Noradrenergic damage can therefore influence the course of the disease [398]. Kinnerup et al. [399] stated that PD rest tremor is linked to noradrenaline. Noradrenaline depletion appears in the thalamus and locus coeruleus of patients without tremor. Two of the most typical features of PD are a loss of noradrenergic neurons in the locus coeruleus (LC) and α-syn pathology. Even though noradrenergic dysfunction is linked to PD non-motor symptoms, a preclinical study indicates that the loss of LC norepinephrine, and, consequently, its immune-modulatory and neuroprotective activities, may exacerbate or even accelerate the progression of PD [400]. 

Other brain regions, such as the midbrain raphe nuclei, also exhibit significant Lewy pathology, which may be a factor contributing to non-motor symptoms. Furthermore, during the premotor stage of PD, there is a disruption of the serotonergic system, which controls mood and emotional pathways [401,402]. Wilson et al. [403] stated that dopaminergic pathology and motor symptoms developed after serotonergic disease were present in premotor A53T *SNCA* carriers and were linked to PD. This emphasizes how serotonergic pathology may have an early role in the development of Parkinson’s disease. Their research showed promise for using serotonin transporter molecular imaging to visualize the premotor pathology of Parkinson’s disease in vivo. By its interactions with receptor proteins, the excitatory neurotransmitter glutamate is responsible for a significant amount of the disruption of normal basal ganglia functioning. Glutamate receptors are linked to the altered neurotransmission in PD and have been shown to have a role in the control of neuronal excitability, transmitter release, and long-term synaptic plasticity [404]. An excess of extracellular glutamate results in abnormal synaptic signaling, which causes excitotoxicity and neuronal death. Furthermore, there is a high correlation between neuroinflammation and glia response with extra synaptic glutamate transport. Glutamate-induced excitotoxicity is primarily associated with glial cells’ compromised capacity to absorb and react to glutamate; this is believed to be a typical hallmark of PD [33,405]. As such, they are seen as novel targets for enhancing the treatment approaches employed in the management of PD [406,407].

## 4. Conclusions

Parkinson’s disease (PD) poses a significant global health challenge, affecting millions of individuals worldwide. Despite extensive research and promising treatment avenues, PD remains incurable, and several aspects of the disease, including its exact etiology and pathogenic mechanisms impacting neurons, remain poorly understood. The multifaceted nature of PD pathogenesis involves a complex interplay of risk factors such as aging, gender, ethnicity, environmental influences, and genetics. The interaction of these factors contributes to the intricate landscape of PD development.

Genetic and molecular mechanisms play pivotal roles in PD, with mutations in key genes—*SNCA*, *LRRK2*, *PINK1*, Parkin, DJ-1, *VPS35*, and *GBA1*—being implicated in the disease. The pathogenic pathways associated with these genetic mutations encompass misfolded protein accumulation, oxidative stress, mitochondrial dysfunction, energy deficits, excitotoxicity, cell-autonomous processes, prion-like characteristics of α-synuclein, and malfunctioning protein clearance pathways.

Cell death emerges as the ultimate outcome of these interconnected pathogenic events, including mitochondrial failure, oxidative and nitrosative stress, and neuroinflammation. Abnormal accumulation of α-synuclein is identified as a central player in PD pathophysiology, triggering cascades of inflammatory processes and heightened neuronal stress. The diverse genetic mutations disrupt normal cellular and molecular mechanisms, giving rise to distinct pathogenic pathways culminating in PD.

Recognizing the substantial overlap between genetics and the molecular mechanisms of PD is crucial. Establishing the link between the genetic basis and associated molecular pathways enhances our comprehension of PD pathology. This knowledge contributes not only to the formulation of preventive treatments and the quest for a cure but also holds the potential for optimizing clinical trial designs and developing improved therapeutic strategies. A personalized approach based on genetic and molecular insights may pave the way for more effective and tailored interventions, minimizing adverse effects and offering hope for the better management of PD.

## Figures and Tables

**Figure 1 biomolecules-14-00073-f001:**
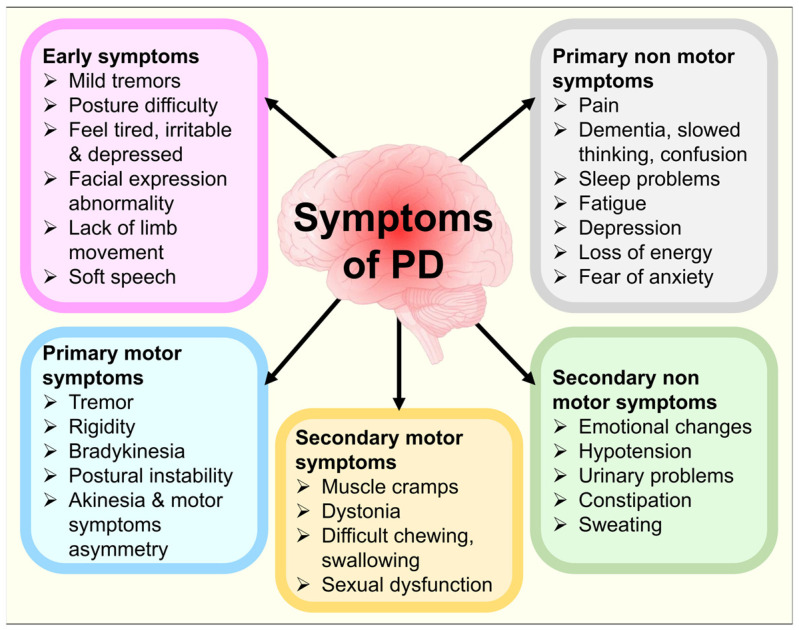
Distinct symptoms of PD [25]. The primary symptoms of PD are mainly categorized into five types: “early symptoms, primary motor symptoms, secondary motor symptoms, primary non-motor symptoms, and secondary non-motor symptoms”.

**Figure 2 biomolecules-14-00073-f002:**
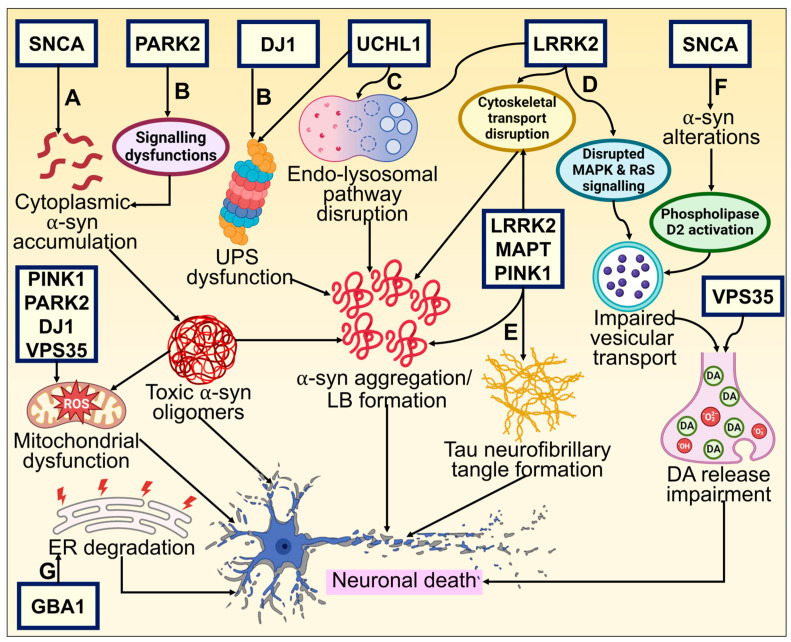
Genetic basis of PD and its underlying molecular pathways toward neurodegeneration. (A) The neuronal defense against α-syn aggregating could consequently be compromised by mutations altering these proteins. The cytoplasmic concentration of the α-syn monomer is increased by missense mutations and chromosomal multiplications of *SNCA*, promoting oligomerization of α-syn that is cytotoxic. This results in neuronal membrane damage and mitochondrial dysfunction [27,94]. (B) DJ-1 and Parkin (encoded by *PARK2*) interact and participate in regular UPS operations [95]. Mutations affecting these proteins may reduce the neuron’s ability to respond to α-synuclein aggregation. α-synuclein aggregates that build up inside neurites and axons in late-onset PD when there is residual UPS function eventually become trapped inside a central Lewy body in surviving neurons. DJ1 also possesses antioxidative characteristics, which may offer an additional connection to α-synuclein fibrillization and impaired function [96]. (C) UCHL1 preserves a pool of monoubiquitin for E3 ligase and UPS function while inhibiting the degradation of free ubiquitin in the endosomal–lysosomal pathway [97,98]. UPS functioning and protein buildup clearance need ATP generation by mitochondria. Loss of *PINK1*, DJ-1, and Parkin activities significantly impairs normal mitochondrial activity, which leads to early-onset parkinsonism [99]. (D, E) Tau (encoded by microtubule-associated protein tau—MAPT) typically maintains the microtubule network in equilibrium, facilitating intracellular signaling and neuronal trafficking. Abnormal phosphorylation impairs its functionality and causes neurofibrillary tangles to develop [100,101]. Phosphorylation, intracellular signaling, and cellular trafficking all seem to be interconnected events requiring *LRRK2* [95]. (F) Mutation-induced altered α-syn activity causes reduced vesicular binding, which negates the inhibition of phospholipase D2, an enzyme implicated in vesicle trafficking and lipid-mediated signaling cascades [102]. Further, restricted release of neurotransmitters and their buildup in the cytosol may produce reactive oxygen species, which in turn causes neuronal death. (G) *GBA1* mutation leads to ER stress activation and degradation, which causes DAergic neuronal death [103].

**Figure 3 biomolecules-14-00073-f003:**
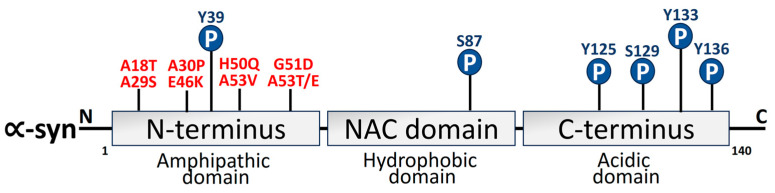
Schematic depiction of α-syn structure domains. Three different domains can be distinguished from the 140-amino-acid α-syn protein. The N-terminus amphipathic domain is composed of KTKEGV repeats that contain the amino acid residues affected by the primary α-syn gene mutations (A30P, E46K, H50Q, G51D, A53T, and A53E) in PD [112,113]. Mutations linked to Parkinson’s disease (red) and phosphorylation sites (blue) have also been depicted.

**Figure 4 biomolecules-14-00073-f004:**
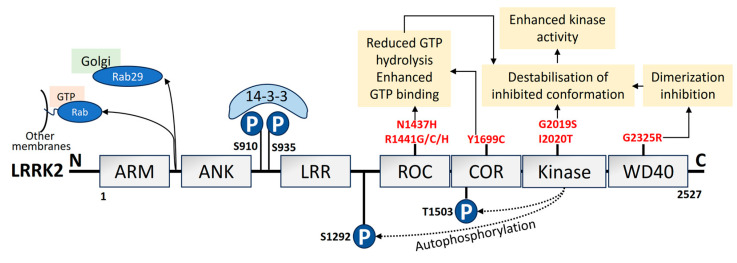
Domain organization, upstream regulation, and PD-linked pathogenic mutations of *LRRK2*. *LRRK2* is made of 2527 amino acids. It consists of 7 domains: ARM, armadillo; ANK, ankyrin; LRR, Leucine-rich repeat; ROC, Ras of complex; COR, C-terminal of ROC; kinase; and WD40 domain [168]. The phosphorylation sites in the N-terminus are pSer910 and pSer935 (blue), which mediate 14-3-3 binding to *LRRK2*. The autophosphorylation sites are pSer1292 and pThr1503 (blue). PD mutations (red) lead to the pathological mechanism that increases kinase activity. Upstream regulation by *LRRK2*, such as the pathogenic *VPS35* mutation, potential *LRRK2* recruitment by unidentified Rabs to other organelle membranes or vesicles, and Rab29 recruitment to the Golgi has also been depicted [169].

**Figure 5 biomolecules-14-00073-f005:**
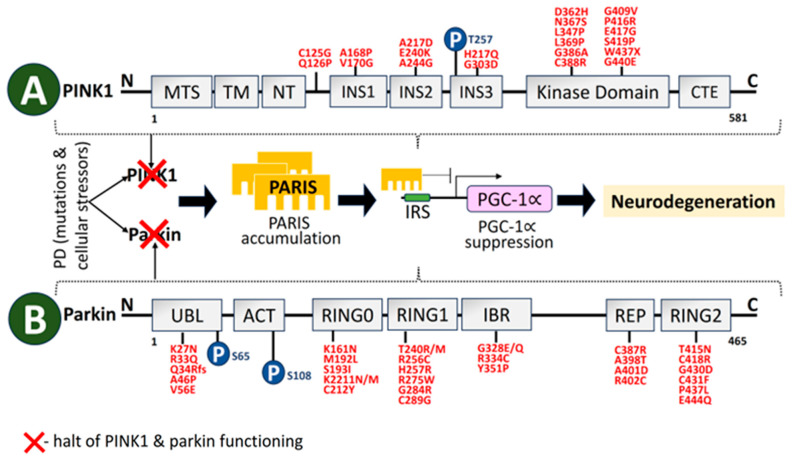
Schematic representation of structural domains of the mitochondrial-associated kinase *PINK1*, and the RBR-E3 Ubiquitin Ligase Parkin. (A) *PINK1* is divided into distinct sections. Individual domains are depicted and labeled as follows: MTS, mitochondrial targeting sequence; TM, transmembrane domain; NT, N-terminal, regulatory domain; INS, insertion; CTE, C-terminal extension [185]. Depending on the residues and protein areas affected, *PINK1*-PD mutations (red) can be classified as having an impact on substrate binding, kinase activity, or *PINK1* structure. (B) Parkin is comprised of 465 amino acids. Individual domains are depicted and labeled as follows: UBL, ubiquitin-like domain; ACT, activating element; RING, really interesting new gene domain; IBR, in-between-RING domain; REP, repressor element [186]. Mutations in *PINK1* and Parkin cause PARIS accumulation that leads to neurodegeneration. Mutations linked to parkinson’s disease (red) and phosphorylation sites (blue) in *PINK1* and parkin have also been depicted.

**Figure 6 biomolecules-14-00073-f006:**
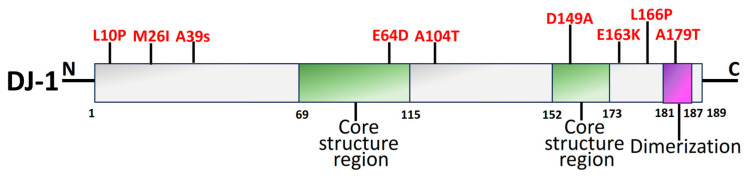
An illustration of functional domains of the DJ-1 protein with pathogenic mutations (red). The two core structural areas (green) and the dimerization region (purple) make up the DJ-1 protein. DJ-1 is a single-domain protein with 189 amino acids [208].

**Figure 7 biomolecules-14-00073-f007:**
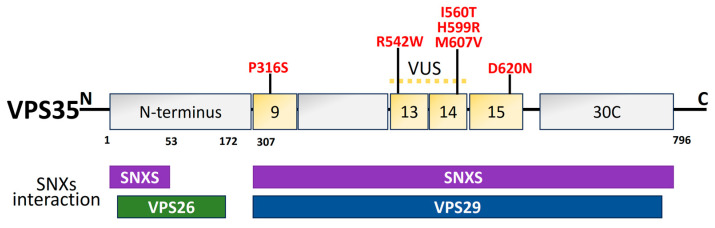
Schematic illustration of *VPS35* structural domains and interactions along with the PD mutations (red) [208]. The dimer of sinexin (SNXs) and *VPS26*, *VPS29*, and *VPS35* combine to form the retromer cargo recognition complex. For the interaction with *VPS26* and *VPS29*, the amino acid residues 1–172 in the N-terminal region and 307–796 in the C-terminal region are significant. The N- and C-terminal regions of the amino acid residues are those that interact with the SNXs. Thirteen of the thirty-four helices projected for a structural level *VPS35*, a right-handed α-helix solenoid, are predicted to be in the C-terminal. In PD, several missense mutations have been discovered. The location of the *VPS35* variation of unknown significance (VUS) is between exons 9 and 14. Exon 15 is the site of the pathogenic mutation (D620N).

**Figure 8 biomolecules-14-00073-f008:**
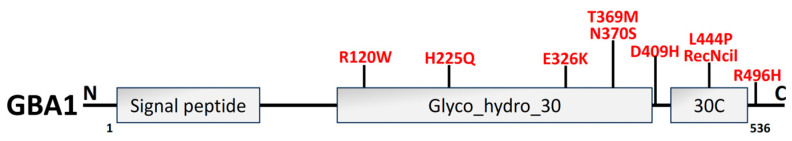
An illustration of *GBA1* structural organization along with PD mutations (red). *GBA1* protein is composed of 497 amino acids, which have three primary domains: 39-residue signal peptide, the conserved catalytic domain Glyco_hydro_30 (329 amino acids), and Glyco_hydro_30C domain (30C; 62 amino acids) [233].

**Figure 9 biomolecules-14-00073-f009:**
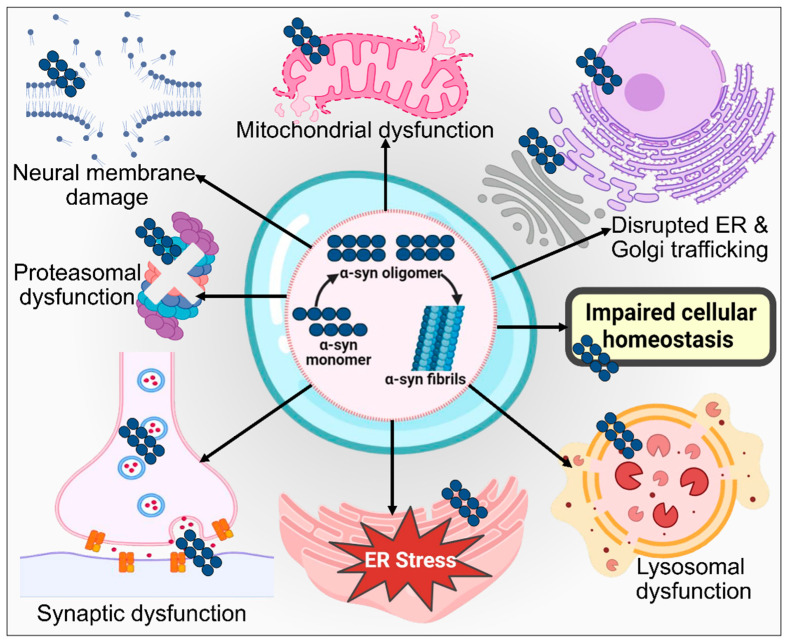
An illustration of the α-syn protein misfolding in the pathogenesis of PD. There are various processes by which α-syn oligomers might cause toxicity. Several protein degradation mechanisms, like autophagy and lysosomal degradation, are adversely affected by oligomers, which also actively contribute to the disruption of mitochondrial function. Additionally, oligomeric α-syn induces ER stress by manipulating the autophagy lysosomal pathways and UPS. α-syn oligomers may hinder ER-Golgi trafficking, axonal transmission, and synaptic impairment by preventing the development of SNARE complexes. Furthermore, by changing the membrane homeostasis, α-syn oligomers can directly trigger cytotoxicity. Ultimately, α-syn oligomers disrupt numerous intracellular signaling pathways and destroy organelles, which may result in neuronal death in PD.

**Figure 10 biomolecules-14-00073-f010:**
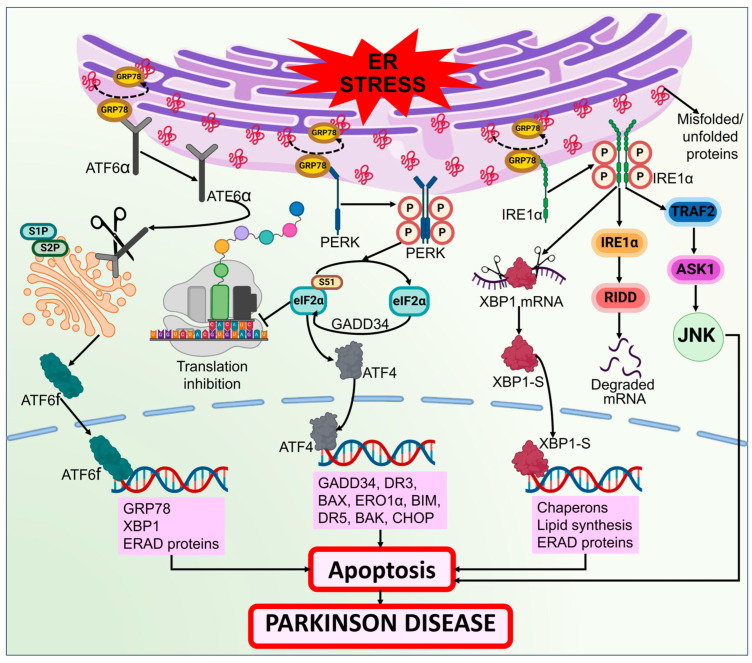
An illustration of ER stress and UPR signaling pathways in the pathogenesis of PD. Stressful circumstances brought on by oxidative stress, infections, nutritional deficiency, and ER alterations in calcium levels can cause protein folding errors to build up in the ER. Three sensor proteins—PERK, IRE1, and ATF6—control UPR by preventing the accumulation of improperly folded proteins and enhancing ER folding ability. GRP78 binds to misfolded proteins like α-syn when the ER is stressed, causing dissociation from GRP78 activates IRE1, ATF6, and PERK. They also activate caspase-3 and force the nucleus to produce CHOP. Consequently, apoptosis occurs in PD. 1-Methyl-4-phenylpyridinium (MPP+) increases the expression of CHOP, which causes ER stress.

**Figure 11 biomolecules-14-00073-f011:**
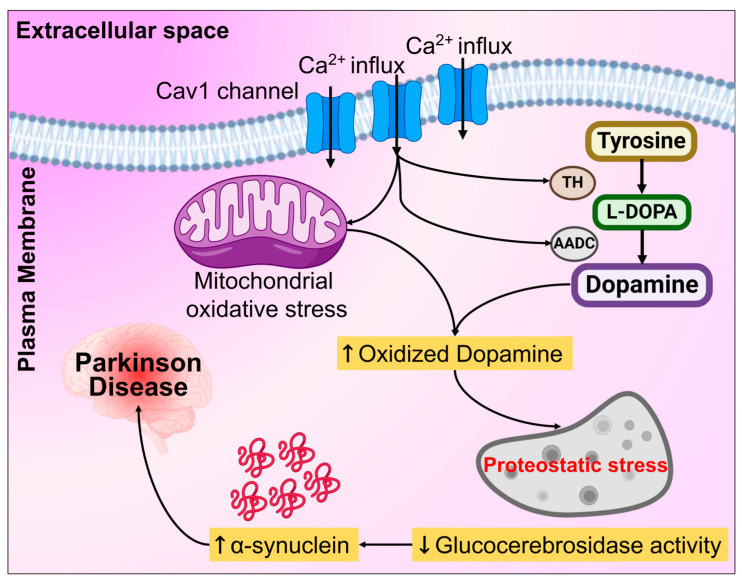
An illustration of DA toxicity in PD pathogenesis. Mitochondrial oxidative stress causes oxidizing DA to build up, which then causes lysosomal failure, decreased glucocerebrosidase enzyme activity, and α-syn deposits in PD neurons. Increased DA production along with impacts on the functioning of mitochondria may be simultaneously caused by higher cytosolic Ca^2+^ concentration through caveolin-1 (Cav1) channels.

**Figure 12 biomolecules-14-00073-f012:**
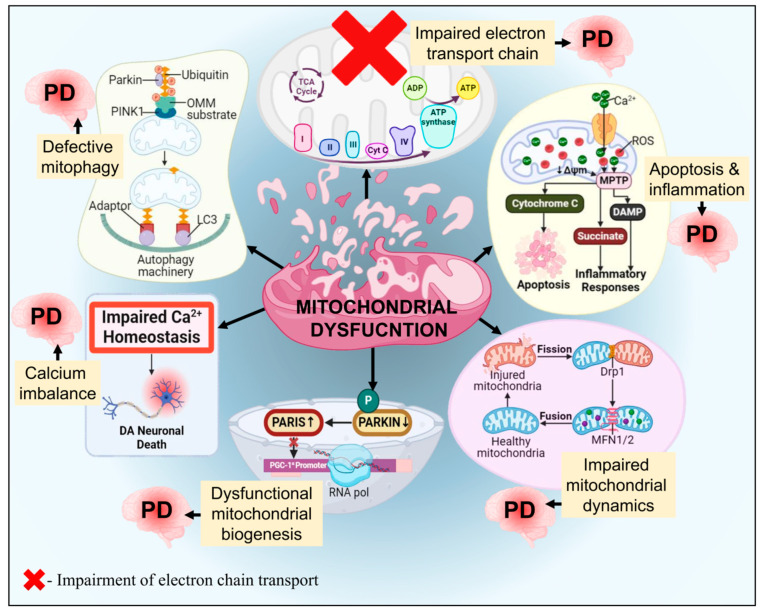
An illustration of the potential cellular mechanism of dysfunctional mitochondria in the pathogenesis of PD. Impaired biogenesis of mitochondria, elevated generation of ROS, impaired mitophagy, compromised trafficking, malfunction of the ECT, deviations in mitochondrial dynamics, calcium imbalance, or combinational processes all contribute to mitochondrial dysfunction linked to the pathogenesis of PD. The possible complicated interaction of the numerous processes leads to a vicious cycle of escalating cellular dysfunction, which in turn causes the neurodegeneration that underpins and accelerates the pathogenesis of PD.

**Figure 13 biomolecules-14-00073-f013:**
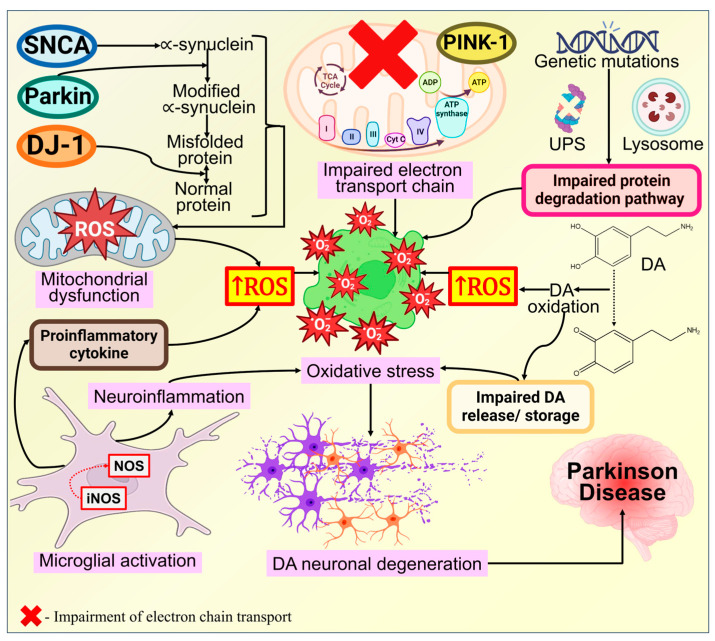
A systematic illustration of oxidative stress-mediated pathogenesis in PD. Several pathways and associated dysfunctions bring on rising oxidative stress because of genetic alterations in PD-related genes. Protein misfolding, mitochondrial damage, and oxidative stress are caused by mutations or modified protein expression. Free radical production and protein aggregation, particularly that of α-syn, are exacerbated by mitochondrial failure. Additionally, the chemical breakdown of DA can contribute to reactive DA quinones, which raise the amounts of ROS. Excessive oxidative stress leads to compromised UPS performance, which in turn is responsible for damaged/ misfolded protein degradation, further compromising cell viability. All of these distinct molecular mechanisms associated with oxidative stress are interlinked in the DAergic neuronal selective degeneration.

**Figure 14 biomolecules-14-00073-f014:**
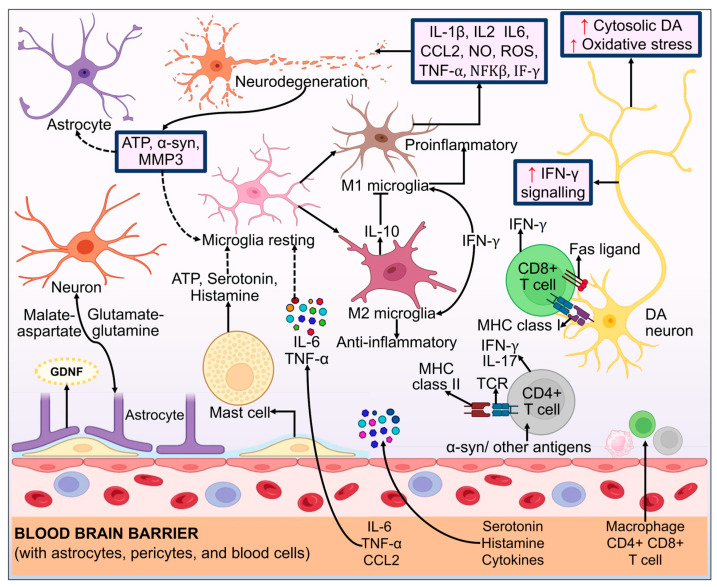
An illustration of neuroinflammation and immune dysregulation in the pathogenesis of PD. The declining strength of the immune response and the interaction between various cell types in the brain contribute to neuroinflammation. α-syn aggregation can interfere with the homeostatic functioning of neurons, astrocytes, microglia, or endothelial cells and can cause a rise in receptor response and the release of chemokines and proinflammatory cytokines. Microglial cells shift from the resting state to the activated M1 microglia, and they release proinflammatory cytokines that aid in the degeneration of DAergic neurons. Furthermore, in cross interactions with astrocytes and microglia, neuronal failure can produce α-syn, ATP, and matrix metalloproteinase-3 (MMP-3), amongst other substances, intensifying the toxic loop of neuroinflammation. Resting microglia are activated to M2 microglia by IL4 and IL13, which then downregulate M1 functionality by releasing IL10 cytokines, which have an anti-inflammatory effect on the CNS. The brain parenchyma is inhabited by CD4+ & CD8+ T cells, and these mediators or the dearth of their effective recovery mechanisms, further exacerbate the proinflammatory state.

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
