# Peer review of "Advancements in Genetic and Biochemical Insights: Unraveling the Etiopathogenesis of Neurodegeneration in Parkinson’s Disease"

_biomolecules, 2024, doi:10.3390/biom14010073_

Round 1

Reviewer 1 Report

Comments and Suggestions for Authors

Strengths:

The review is broad and addresses many molecular aspects of PD.

Extensive graphical summaries.

Novel literature cited showing latest research.

The text is written in good language although some strange choice of word use was noted.

Weakness:

The text is overpacked with data and chaotic. Many repetitions in the text.

Text has structure flaws, mixes molecular and behavioural symptoms, genetic with sporadic.

There is missing a clear division what is only suggested as a mechanism based on experimental models and what is proven in human. Strong statements - potentially misleading. Needs to be carefully revisioned.

Lacs perspective and helpful narration. Many interesting information gathered together. All shown as a single point of view, while there are many discrepancies and many variations of the processes described in the different studies which should be pointed out. Everything seems to be important in the same way, which is not.

What information is still missing but essential for future PD research – this should be also indicated as a perspective. 

Specific issues:

Abstract:

-        Should include more detailed info about the content of the text.

-        Non-motor as well as non-dopaminergic pathologies should be indicated.

Intorduction

Line 42 – “this one” refers to the disease or one of the symptoms?

Line 48 -  “PD is a global disease that affects 1-2% of people over 65” – this is a repetition

Line 50-51 “domestic instances of PD”? not comprehensive

Line 57 – “striatum is selectively degenerated” - non-dopaminergic pathologies should be also indicated.

Line 57 – “15 to 20 years” – please provide original references for those numbers.

Line 63 – Lewy bodies are also present in other than SNc brain structures. They are also markers of synucleinopathies.

The whole text: striatum is a rodent brain structure. When discussing human brain please refer to as caudate-putamen.

Fig.1 – how come sexual dysfunction is a motor symptom? Akinesia and motor symptoms asymmetry are missing.

Line 73 and 75 – “folding errors in proteins” ? it is not precise enough statement. Rather misfolding or abnormal folding.

Line 74 – “energy collapse”?

Line 74 – “prion like protein infection” – no infection was ever proven. Rather prion-like abilities of alpha synuclein.

Line 77 – tau and amyloid beta are not the main proteins involved in LB. See also fig 2 description.

Line 80 – “The intracellular aggregated -syn tears a hole in the neuronal membrane, leading to neuronal death [24]” – oversimplifying and not proven.

Lines 80 – 85 – molecular and behavioural aspects are mixed together – please sequester in separate paragraphs.

Line 88 – “As  a  result  of  the  impaired  ubiquitin-proteasome  system (UPS) and mitochondrial damage, gene mutations occur” – not true.

 Line 142-147 – synuclein functions, abnormal aggregation, fibrils and oligomers formation, interaction with lipids and vesicles should be described first as a main issue in PD, not orexin or spectrin, which should be indicated later.

Line 151 – “be indulged in synaptic transmission” – indulged?

Line 186 – “A53T causes apoptotic pathways in adrenal phaeochromocytoma (PC12) cells” – please narrate clearly what is defined and proven in human PD condition and what is shown in experimental models and only suggested for mechanisms in human brain. There is a mix-up of observed facts from different experimental studies that have nothing to do with the human disease itself. In vitro observation is not a proof for human brain mechanism. Clearly define what is suggested and what proven, what shown in human and what in experimental studies.

For example “DAergic neurons die as a result of -syn selective binding to tropomyosin receptor kinase B (TrkB) and inhibition of the TrkB signaling pathway [69,70].” This is experimental observation, not proven in human. But it is narrated as a definitive proven fact. Please mind this subtle issue throughout the whole text and comment observations accordingly. Writing a review has a responsibility to put a proper perspective on different facts, not only to put them altogether.

Line 203 – “Age affects how well the penetrance works” – I don’t know what authors ment… in the aspect of PD.

 Creating a list of genes causing PD vs increasing susceptibility would be useful.

 Line 236 - IMM - no abbreviation definition.

Line 274 – “DJ-1 (PARK7), is a progressive old gene” what does it mean?

Line 315 – “the catabolization” or catabolism

Line 344 – in which animal model ? why hippocampus and serotonin would be important in that story? It was not mentioned before. Need to explain to the reader

Line 347 – is the c-Abl kinase the only way to phosphorylate synuclein? Please add perspective to this paragraph.

Line 359 – are pathogenes the only agents removed by autophagy and mitophagy? Please correct.

Line 392 – “DAergic neurons die due to ER stress” this is example of a very strict statement that seems the only one and proven. But ER stress is not the only cause of neurodegeneration in PD patients. Authors need to soften such statements and put more realistic perspective to the whole text.

Line 408 – “The peripheral inflammation in PD is exacerbated by lymph node swelling” – is there evidence of peripheral inflammation and lymph nodes swelling in PD patients related to the PD progression?

Line 419 – what are exactly “clinical models of PD”?

Line 421 – what is CHOP abbreviated for?

Paragraph 3.3 calcium homeostasis – it is quite vague description. How excitotoxicity and MPTP fits in that story?

Line 461 – “that maintains the Parkinson's tremor” – not only tremor.

Lines 461 and 465 – authors write on DA decline and then conclude on elevated DA. – this whole paragraph should be better commented/narrated, not only to show random facts together.

Paragraph 3.5 mitochondria normally produce ROS, which are also necessary signalling molecules. There are numerous safety switches in mitochondria that do not allow them to decrease ATP synthesis even when they are “defective”. Therefore “disruption of mitochondria functioning” is an exaggeration.

There are multiple issues with mitochondria in PD, why authors chose HtrA2 mechanism as the most important? Please justify.

Line 525 and 528 – “MPP+ is regarded as a major histocompatibility complex type I (MHCI) inhibitor” – ? or did you mean mitochondrial complex I?

Line 549 – ROS does not build up – it reacts too quickly.

Lines 550-563 – all those information are very loosely fitting to each other. Lipid pathways are important but that connects them to oxidative stress?

Lines 620-627 – repetitions

Figure 8 description – there is no LPS in the human PD brain. Therefore description “An illustration of neuroinflammation and immune dysregulation in PD” is inadequate. LPS and IFNgamma are research tools. This is an example of mixing processes really involved in human PD with experimental results, hence suggestions of processes. This is potentially misleading for the reader. Throughout the whole text it has to be carefully discriminated what information comes from experimental models and what was proven in human brain.

Lines 652 – M1 and M2 phenotypes are not recognised anymore as a realistic description of microglia phenotype. Rather activation spectrum is approved.

Paragraph 4.1 Aging – repetition

Line 698 – do not ‘blame’ anything in scientific text.

Paragraph 4.4 genetics – repetition

4.5 – “certain environmental factors”, “some metals” – please be more specific. These are empty words. Its better to write shorter text but more meaningful.

Paragraph 4. Risk factors associated with PD – This part is superficial and incomplete. This should be either implemented within the Introduction or be cut out.

Line 741 – cytology? Please rephrase

Those are not conclusions:

Line 742 - “causes of PD have been well recognized to be complicated” ; “Researchers continue to study the possible ways” ; “a lot remains unknown and poorly understood about aspects”…

 Authors conclude on potential therapy while the whole text was on molecular and genetic aspects. Please summarize and conclude on your subject.

Comments on the Quality of English Language

The text is written in good language although some strange choice of word use was noted.

Author Response

Reviewers’ Comments to the Authors:

Responses to Reviewer 1 Comments

S. No.

Comments

Responses

REVIEWER 1

1

Strengths:

The review is broad and addresses many molecular aspects of PD.

Extensive graphical summaries.

Novel literature cited showing latest research.

The text is written in good language although some strange choice of word use was noted.

The authors appreciate the reviewer for their remarks.

2

Weakness:

·         The text is overpacked with data and chaotic. Many repetitions in the text.

·         Text has structure flaws, mixes molecular and behavioural symptoms, genetic with sporadic.

·         There is missing a clear division what is only suggested as a mechanism based on experimental models and what is proven in human. Strong statements - potentially misleading. Needs to be carefully revisioned.

·         Lacs perspective and helpful narration. Many interesting information gathered together. All shown as a single point of view, while there are many discrepancies and many variations of the processes described in the different studies which should be pointed out. Everything seems to be important in the same way, which is not.

·         What information is still missing but essential for future PD research– this should be also indicated as a perspective.

·         The manuscript has been thoroughly checked and structural changes and corrections have been made at various places of the manuscript to improve its readability and understanding. Furthermore, repetitions have also been omitted in the manuscript and highlighted using the “track changes” function.

·         As suggested, the structural changes have been made at various places in the manuscript after thoroughly reviewing it, and corrections are highlighted using the “track changes” function at respective places. The detailed response to this specific query can be found in issue number 16, as raised by reviewer 1.

·         In accordance with the suggestion, the entire manuscript has undergone a thorough review and correction process to delineate the research findings distinctly. The detailed response to this specific query can be found in issue numbers 20 and 41, as raised by reviewer 1.

·         Thanks for pointing this out. As suggested, we have corrected all such issues in the manuscript, and detailed responses to these specific queries are covered under issue numbers 20, 26, 28, and 34 raised by reviewer 1.

·         In response to the reviewer's suggestion, the conclusion section now includes added lines on the future perspective of Parkinson's disease, specifically addressing the subject matter of the present study. Please refer to lines 1105-1132.

Specific Issues

3

Abstract:

-Should include more detailed info about the content of the text.

In response to the reviewer’s suggestion, additional details concerning the content discussed in the manuscript have been integrated into the abstract. The restructuring of the abstract, while adhering to the 200-word limit, involved the omission of certain lines to enhance clarity and focus. Please refer to page 1, lines 20-34 in the revised file for specific changes.

-Non-motor as well as non-dopaminergic pathologies should be indicated.

-Non-motor and non-dopaminergic pathologies in PD have been dedicatedly added and discussed in sections 3.11 and 3.12 on pages 34-37. It has also been stated in the abstract.

4

Introduction

Line 42 - “this one” refers to the disease or one of the symptoms?

The term “this one” referring to the disease has been substituted with “PD” to alleviate any potential confusion for the readers. Kindly review page 2, line 48, in the revised document for this modification.

5

Line 48 - “PD is a global disease that affects 1-2% of people over 65” – this is a repetition

The repetition in line 45 has been eliminated and deliberately retained in line 55. Please refer to the revised file for verification.

6

Line 50-51 “domestic instances of PD”? not comprehensive

The identified issue has been rectified, and the phrase “domestic instances of PD” has been replaced with “familial instances of PD.” Please review line 58 in the revised document for confirmation.

7

Line 57 – “striatum is selectively degenerated” - non-dopaminergic pathologies should be also indicated.

Following the reviewer’s suggestion, non-dopaminergic pathologies in PD have been introduced in lines 64-74. Furthermore, these non-dopaminergic pathologies in PD have been thoroughly addressed and discussed in sections 3.11 and 3.12.

8

Line 57 – “15 to 20 years” – please provide original references for those numbers.

The correct reference for the mentioned line has been provided as [17]. Please refer to lines 75-76 for verification.

9

Line 63 – Lewy bodies are also present in other than SNc brain structures. They are also markers of synucleinopathies.

The whole text: striatum is a rodent brain structure. When discussing human brain please refer to as caudate-putamen.

The term “striatum” has been verified and cross-checked in the manuscript. Appropriate corrections have been made as needed. Please refer to line 723 for confirmation.

10

Fig.1 – how come sexual dysfunction is a motor symptom? Akinesia and motor symptoms asymmetry are missing.

The error has been rectified in Fig. 1. Sexual dysfunction has been appropriately placed under non-motor symptoms, and akinesia and motor symptom asymmetry have been added to the Figure for accuracy.

11

Line 73 and 75 – “folding errors in proteins”? it is not precise enough statement. Rather misfolding or abnormal folding.

In agreement with the reviewer’s comment, “folding errors in proteins” have been corrected by replacing the term with “misfolding” and “abnormal folding” to avoid reader confusion. Please refer to page 4, lines 95 and 99.

12

Line 74 – “energy collapse”?

The “energy collapse” refers to the influence of energy deficiency and energy metabolism on the subcellular level of SNPC in PD pathogenesis. For a better understanding, the term “energy collapse” has been replaced with “energy deficiency.” Please refer to page 4, line 96.

13

Line 74 – “prion like protein infection” – no infection was ever proven. Rather prion-like abilities of alpha synuclein.

The necessary correction has been implemented. Kindly refer to page 4, line 97, for confirmation.

14

Line 77 – tau and amyloid beta are not the main proteins involved in LB. See also Fig 2 description.

The ‘LB’ are the main misfolded protein inclusion found in the intracellular spaces of SNPC in PD. These bodies comprise a variety of misfolded proteins such as ubiquitin (Ub) and α-syn, phosphorylated tau (p-tau), and amyloid-β.

The sentence has been revised and corrected. Please refer to page 4, lines 102-103, for the updated version.

Additionally, the caption of Figure 2 has been rewritten for clarity. Please review the revised caption for accurate information.

15

Line 80 – “The intracellular aggregated a-syn tears a hole in the neuronal membrane, leading to neuronal death [24]” – oversimplifying and not proven.

The sentence has been clarified and corrected to read as follows: “The intracellular aggregated α-syn permeates the cell membrane, leading to neuronal death.” Please refer to page 4, lines 104, for the revised version.

16

Lines 80 – 85 – molecular and behavioural aspects are mixed together – please sequester in separate paragraphs.

The lines pertaining to the behavioral aspects (lines 105-106) have been relocated to the appropriate paragraph. Refer to page 2, lines 75-76, for the adjusted placement.

17

Line 88 – “As  a  result  of  the  impaired  ubiquitin-proteasome  system (UPS) and mitochondrial damage, gene mutations occur” – not true.

Acknowledging the reviewer’s feedback, the highlighted sentences have been omitted. Please refer to page 4, lines 133-136, for the revised version.

18

 Line 142-147 – synuclein functions, abnormal aggregation, fibrils and oligomers formation, interaction with lipids and vesicles should be described first as a main issue in PD, not orexin or spectrin, which should be indicated later.

The sequencing issue has been rectified, and the lines (274-282) have been appropriately shifted to page 9, lines 311-319. Please verify the revised placement.

19

Line 151 – “be indulged in synaptic transmission” – indulged?              

The typographical error has been rectified, and the correct term ‘involved’ has been used. Please refer to line 289 for verification.

20

Line 186 – “A53T causes apoptotic pathways in adrenal phaeochromocytoma (PC12) cells” – please narrate clearly what is defined and proven in human PD condition and what is shown in experimental models and only suggested for mechanisms in human brain. There is a mix-up of observed facts from different experimental studies that have nothing to do with the human disease itself. In vitro observation is not a proof for human brain mechanism. Clearly define what is suggested and what proven, what shown in human and what in experimental studies.

For example “DAergic neurons die as a result of a-syn selective binding to tropomyosin receptor kinase B (TrkB) and inhibition of the TrkB signaling pathway [69,70].” This is experimental observation, not proven in human. But it is narrated as a definitive proven fact. Please mind this subtle issue throughout the whole text and comment observations accordingly. Writing a review has a responsibility to put a proper perspective on different facts, not only to put them altogether.

The investigations involving experimental models, the human brain, and in-vitro studies have been meticulously delineated and presented distinctly throughout the manuscript, as indicated in multiple instances, including references to lines 332-350, 606-620, 647-665, 727-747, 966-984 and various other places.

21

Line 203 – “Age affects how well the penetrance works” – I don’t know what authors ment… in the aspect of PD.

 Creating a list of genes causing PD vs increasing susceptibility would be useful.

The inadvertently included line has been removed. Please verify the changes starting from line 353 for confirmation.

22

Line 236 - IMM - no abbreviation definition.

The abbreviation “IMM” has been replaced with its full form, “inner mitochondrial membrane,” as it once appeared. Please refer to line 408.

23

Line 274 – “DJ-1 (PARK7), is a progressive old gene” what does it mean?

This suggests that DJ-1 is a gene with ancient evolutionary origins. The term “progressive old gene” has been replaced with “evolutionary ancient gene,” as specified in line 469, to improve reader understanding.

24

Line 315 – “the catabolization” or catabolism

The correction has been implemented, and the term is now ‘catabolism.’ Please refer to line 528 for confirmation.

25

Line 344 – in which animal model? why hippocampus and serotonin would be important in that story? It was not mentioned before. Need to explain to the reader

The rat model has been explicitly mentioned, as indicated in line 561. In response to the reviewer’s suggestion, the significance of serotonin in hippocampal neurogenesis has been incorporated in lines 562-565. Please review the specified lines for the added information.

26

Line 347 – is the c-Abl kinase the only way to phosphorylate synuclein? Please add perspective to this paragraph.

Certainly, the additional ways to phosphorylate α-synuclein have been addressed. It is clarified that c-Abl kinase is not the exclusive mediator, and other kinases such as G-protein-coupled receptor kinases (GRKs), casein kinase II, polo-like kinases, and leucine-rich repeat kinase 2 (LRRK2) are also involved in phosphorylating α-synuclein Ser129. Please refer to lines 573-575 for the expanded information.

27

Line 359 – are pathogenes the only agents removed by autophagy and mitophagy? Please correct.

The necessary corrections have been implemented. It has been clarified that pathogens are not the sole agents removed by autophagy and mitophagy. Autophagy also facilitates the elimination of damaged organelles and protein aggregates, while mitophagy specifically removes damaged mitochondria. This information has been incorporated into lines 583-585.

28

Line 392 – “DAergic neurons die due to ER stress” this is example of a very strict statement that seems the only one and proven. But ER stress is not the only cause of neurodegeneration in PD patients. Authors need to soften such statements and put more realistic perspective to the whole text.

In response to the reviewer’s suggestion, the sentence has been adjusted in a more nuanced manner. Please refer to lines 622-623 for the modified statement.

“The observed death of dopaminergic (DAergic) neurons in the substantia nigra pars compacta (SNpc) of PD patients has also been linked to ER stress.”

29

Line 408 – “The peripheral inflammation in PD is exacerbated by lymph node swelling” – is there evidence of peripheral inflammation and lymph nodes swelling in PD patients related to the PD progression?

Indeed, certain studies have proposed that inflammation, both within the central nervous system (CNS) and in peripheral tissues, might be involved in the pathogenesis and progression of PD. Peripheral inflammation and the activation of the immune system could potentially contribute to neuroinflammation in the brain, impacting the progression of PD. Please review line 642 for the corresponding information.

Reference:

Muñoz-Delgado, L., Labrador-Espinosa, M. Á., Macías-García, D., et al. (2023). Peripheral Inflammation is Associated with Dopaminergic Degeneration in Parkinson’s Disease. Movement disorders, 38(5), 755–763. https://doi.org/10.1002/mds.29369

30

Line 419 – what are exactly “clinical models of PD”?

A typographical error was identified; the authors meant to convey “preclinical studies (rat and mouse).” The necessary correction has been made. Please refer to line 653 for accurate information.

31

Line 421 – what is CHOP abbreviated for?

The full form for the abbreviation “CHOP” has been mentioned. Please refer to line 655.

32

Paragraph 3.3 calcium homeostasis – it is quite vague description. How excitotoxicity and MPTP fits in that story?

Paragraph 3.3, emphasizing ‘calcium homeostasis,’ has been rephrased to enhance clarity, removing misfit sentences. Please refer to lines 676-701 for the revised version.

33

Line 461 – “that maintains the Parkinson’s tremor” – not only tremor.

The authors express gratitude to the reviewer for bringing this to attention. In addition to tremors, dopamine cells also play a role in maintaining rigidity, impaired balance, and slowed movement and coordination. This information has been added to lines 723-724 for clarification.

34

Lines 461 and 465 – authors write on DA decline and then conclude on elevated DA. – this whole paragraph should be better commented/narrated, not only to show random facts together.

The corrections have been made, and the lines have been rewritten with correct information. Please refer to lines 722-747.

35

Paragraph 3.5 mitochondria normally produce ROS, which are also necessary signalling molecules. There are numerous safety switches in mitochondria that do not allow them to decrease ATP synthesis even when they are “defective”. Therefore “disruption of mitochondria functioning” is an exaggeration.

The phrase “disruption of mitochondria functioning” has been revised and replaced with “mitochondrial dysfunctioning” for improved clarity. Please refer to line 761-762 for the updated wording.

36

There are multiple issues with mitochondria in PD, why authors chose HtrA2 mechanism as the most important? Please justify.

The paragraph addresses several crucial issues, including Alpha-synuclein oligomers, alpha-synuclein toxicity, and lipid peroxidation that contribute to alpha-synuclein toxicity. These factors are identified as causes leading to mitochondrial dysfunction in PD. However, it’s important to note that a recent in vitro and preclinical discovery highlighting the role of HtrA2 in PD pathogenesis was also mentioned, pending clinical validation.

The sentence has been corrected to enhance its clarity and meaning. Please refer to lines 772-777 for the revised version.

37

Line 525 and 528 – “MPP+ is regarded as a major histocompatibility complex type I (MHCI) inhibitor” – ? or did you mean mitochondrial complex I?

The correction has been made, and it is now ‘mitochondrial complex I.’ Please refer to lines 796 and 799 for confirmation.

38

Line 549 – ROS does not build up – it reacts too quickly.

Corrections have been made. Please refer to lines 820-821.

39

Lines 550-563 – all those information are very loosely fitting to each other. Lipid pathways are important but that connects them to oxidative stress?

Certainly, the paragraph has been restructured and rephrased for improved reader understanding. Please refer to lines 810-838.

Lipid pathways play a pivotal role in the context of oxidative stress, engaging through various mechanisms such as lipid peroxidation, the formation of oxidized lipids, lipid-derived free radicals, and more. The susceptibility of fatty acids to lipid peroxidation, a consequence of free radical-mediated injury in PD, is particularly noteworthy due to the high unsaturation of these acids. In instances of oxidative stress, fatty acids become prone to lipid peroxidation, contributing to cellular damage. Moreover, oxidized lipids, proteins, and DNA are released by lipid-dead neurons, initiating a neurotoxic cycle by activating microglia—a detrimental loop in Parkinson’s disease (PD). This cascade forms a vicious cycle of neurotoxicity, exacerbating the impact of oxidative stress on neuronal health. The exposure of certain lipid molecules to oxidative stress can also prompt the generation of lipid-derived free radicals, further amplifying the oxidative stress burden in the context of PD.

40

Lines 620-627 – repetitions

Certainly, the paragraph has been restructured with the removal of repetition and redundant information. Please refer to lines 922-930 for the revised version.

41

Figure 8 description – there is no LPS in the human PD brain. Therefore description “An illustration of neuroinflammation and immune dysregulation in PD” is inadequate. LPS and IFNgamma are research tools. This is an example of mixing processes really involved in human PD with experimental results, hence suggestions of processes. This is potentially misleading for the reader. Throughout the whole text it has to be carefully discriminated what information comes from experimental models and what was proven in human brain.

The description of Figure 8, now numbered as Figure 14, has been rewritten to enhance its understanding. Please refer to the updated description in the corresponding section for the improved version.

42

Lines 652 – M1 and M2 phenotypes are not recognised anymore as a realistic description of microglia phenotype. Rather activation spectrum is approved.

As recommended by the reviewer, the sentence has been revised by employing the activation spectrum of microglia instead of using M1 and M2 phenotypes. Please refer to lines 952-958 for the corrected version.

43

Paragraph 4.1 Aging – repetition

Paragraph 4.1, now mentioned as Paragraph 1.1.1, has been reviewed, and duplication has been omitted. Please refer to lines 148-149 for the revised and improved version.

44

Line 698 – do not ‘blame’ anything in scientific text.

The correction has been implemented by replacing the term ‘blame’ with ‘to be the cause’. Please review line 162 for accurate wording.

45

Paragraph 4.4 genetics – repetition

Paragraph 4.4, now mentioned as Paragraph 1.1.4, has been reviewed, and duplication has been omitted. Please refer to lines 177-178 for the revised and improved version.

46

4.5 – “certain environmental factors”, “some metals” – please be more specific. These are empty words. Its better to write shorter text but more meaningful.

The sentence has been enhanced and corrected as per the reviewer’s suggestion. Please refer to lines 190-191 and 199 for the improved version.

47

Paragraph 4. Risk factors associated with PD – This part is superficial and incomplete. This should be either implemented within the introduction or be cut out.

As suggested, the paragraph discussing “Risk factors associated with PD” has been incorporated within the introduction under subheading 1.1. It is briefly addressed in the manuscript, acknowledging that the primary focus of the writing centers around comprehending the molecular and genetics-based pathogenesis underlying neurodegeneration in PD.

48

Line 741 – cytology? Please rephrase

The conclusion has been restructured and summarized to align with the subject of the writing. The term “cytology” no longer exists in the conclusion. Please review the conclusion section for the updated version.

49

Those are not conclusions:

Line 742 - “causes of PD have been well recognized to be complicated”; “Researchers continue to study the possible ways”; “a lot remains unknown and poorly understood about aspects”…

Authors conclude on potential therapy while the whole text was on molecular and genetic aspects. Please summarize and conclude on your subject.

The conclusion has been corrected and summarized in line with the subject of the writing, considering all the suggestions made by the reviewer. Please refer to the conclusion section for the updated version.

Reviewer 2 Report

Comments and Suggestions for Authors

The article “Advances in Genetic and Biochemical Insights: Unravel-2 ing the Etiopathogenesis of Neurodegeneration in Parkinson’s 3 Disease” attempts to review the current state of research into neurodegenerative Parkinson’s disease. The following are the main comments that arose during the review process of this work:

1. In the Introduction, sources 3, 4, 6, 15 seem quite outdated. More recent data (last 5 years) must be provided.

2. The authors point out rather outdated data, proposing the hypothesis that “the occurrence and development of PD may be influenced by protein folding errors and their accumulation, oxidative stress, mitochondrial dysfunction, energy collapse, excitotoxicity, cell-autonomous processes, and prion-like protein infection” (references 9,19,20). It is necessary to present more modern ideas and theoretical premises to justify PD. In general, the section on the causes of PD looks unconvincing.

3. The data in references (26 and 27) also looks outdated.

4. In the section “Genetic basis of PD”, Scheme 2 is presented, which is overloaded with a large number of symbols, but is not well explained. It is recommended to either reduce the number of symbols on the diagram, or provide more complete and reasoned explanations, supported by references to modern literature.

5. The characterization of α-synuclein is not convincing enough. The authors provide data on the chemical structure of α-synuclein and write that “due to its hydrophobicity, the main hydrophobic region can easily form a β-pleated sheet. The positively charged amino terminus is vulnerable to changes in acetylation and ubiquitination [38,39]” Reference 38 is from 2024 and it is unclear how this reference relates to the 2023 review paper. The authors further write, “α-syn binds to the orexin 1 receptor (OX1R), which facilitates post-translational degradation of the OX1R protein by lysosomal and proteasomal pathways.” This proposal needs to be supported by reference to the literature.

6. In general, in contrast to the well-known effects that cause PD and are presented in Scheme 2, data on the chemical structure of α-synuclein, as well as other proteins (sections 2.2-2.6) and the forms of their existence and metabolism in cells need to be generalized, and possibly a schematic representation. The authors, who presented in a review article a large number of diagrams demonstrating well-known manifestations of PD, could summarize in the form of a diagram the spatial organization and participation of α-synuclein (pp. 134-182) and other proteins in the pathogenesis of PD. This would be a good contribution to the understanding of the pathogenesis of PD, with an attempt to understand the involvement of several proteins in the development of this form of neurodegenerative disease. References 58, 61-64, 66, 72, 73 need to be replaced with more up-to-date data.

7. In section 2.2, the information needs more detailed elaboration; sources 77, 83, 93-97, 101 are outdated, need to be checked

8. Reference 146, 170, 194, 195, needs clarification, no title of the work.

9. There are many outdated references in the work, 173, 177, 178, 187, 194, 195, etc. more up-to-date information needs to be used.

10. The review contains an excessive number of diagrams showing well-known processes associated with the pathogenesis of PD. It is recommended to reduce the number of schemes and make them more general.

11. Stylistic elaboration of the text is necessary; in many places the text of the review article looks raw and unedited.

12. English is not clear everywhere. Significant stylistic correction is required.

Comments on the Quality of English Language

English is not clear everywhere. Significant stylistic correction is required

Author Response

Responses to Reviewer 2 Comments

S. No.

Comments

Responses

REVIEWER 2

1

In the introduction, sources 3, 4, 6, 15 seem quite outdated. More recent data (last 5 years) must be provided.

In the Introduction section, we have updated the references in the first paragraph to include the most recent sources available within the context. Please review the revised content for accuracy and relevance.

2

The authors point out rather outdated data, proposing the hypothesis that “the occurrence and development of PD may be influenced by protein folding errors and their accumulation, oxidative stress, mitochondrial dysfunction, energy collapse, excitotoxicity, cell-autonomous processes, and prion-like protein infection” (references 9,19,20). It is necessary to present more modern ideas and theoretical premises to justify PD. In general, the section on the causes of PD looks unconvincing.

In response to the reviewer’s suggestions, we have incorporated recent ideas and theories on the occurrence and development of Parkinson’s disease, citing the relevant sources. Older references (9, 19, 20) have been replaced with more up-to-date ones. Certain lines within the specified range (99-304) have been restructured to enhance engagement and comprehension. Please review this section for the updated and improved content.

3

The data in references (26 and 27) also looks outdated.

The authors appreciate the reviewer for pointing this out. References have been replaced by the most recent and relevant studies available. Please refer to the relevant section on page 4.

4

In the section “Genetic basis of PD”, Scheme 2 is presented, which is overloaded with a large number of symbols, but is not well explained. It is recommended to either reduce the number of symbols on the diagram, or provide more complete and reasoned explanations, supported by references to modern literature.

In order to enhance the conceptual understanding, Figure 2, titled “Genetic basis of PD,” has been reformed, now subdivided into 7 sub-sections (A, B, C, D, E, F, G).

Responding to the reviewer’s feedback, the caption of Figure 2 has been enriched with more detailed information to thoroughly explain the Figure.

Recent references have also been included to support the reasoned explanations within the caption.

Additionally, as per the suggestion, certain symbols from Figure 2 have been omitted to eliminate ambiguity. Please refer to the revised Figure 2 and its caption for these improvements.

5

The characterization of α-synuclein is not convincing enough. The authors provide data on the chemical structure of α-synuclein and write that “due to its hydrophobicity, the main hydrophobic region can easily form a β-pleated sheet. The positively charged amino terminus is vulnerable to changes in acetylation and ubiquitination [38,39]” Reference 38 is from 2024 and it is unclear how this reference relates to the 2023 review paper. The authors further write, “α-syn binds to the orexin 1 receptor (OX1R), which facilitates post-translational degradation of the OX1R protein by lysosomal and proteasomal pathways.” This proposal needs to be supported by reference to the literature.

Reference 38, now numbered as [111] in line 274, is from 2024, and it’s worth noting that the preprint is already available online ahead of its reprint, with a volume assigned for 2024. This aligns with the common practice of many online open-access journals. The paper referenced is easily accessible via the following link: https://www.cjter.com/EN/abstract/abstract19028.shtml.

For clarity, this information is duly mentioned in the reference section.

Responding to the reviewer’s request for a supportive citation related to the proposal concerning the OX1R receptor, citations from modern literature ([129] and [130]) have been included.

In consideration of another reviewer’s suggestion regarding the better position of the paragraph, the paragraph on orexin and spectrin has been rearranged (lines 311-319).

6

In general, in contrast to the well-known effects that cause PD and are presented in Scheme 2, data on the chemical structure of α-synuclein, as well as other proteins (sections 2.2-2.6) and the forms of their existence and metabolism in cells need to be generalized, and possibly a schematic representation.

The authors, who presented in a review article a large number of diagrams demonstrating well-known manifestations of PD, could summarize in the form of a diagram the spatial organization and participation of α-synuclein (pp. 134-182) and other proteins in the pathogenesis of PD. This would be a good contribution to the understanding of the pathogenesis of PD, with an attempt to understand the involvement of several proteins in the development of this form of neurodegenerative disease.

References 58, 61-64, 66, 72, 73 need to be replaced with more up-to-date data.

The authors thank the reviewer for this valuable and constructive suggestion. Following this guidance, the schematic representations (Figures 3, 4, 5, 6, 7, 8) have been updated to provide a general overview of the structural domains of α-synuclein, metabolism as much as possible, as well as other proteins (sections 2.2-2.6) and the various forms of their existence in cells.

References 58, 61-64, 66, 72, and 73 have been replaced and cited with the most recent literature on the subject.

7

In section 2.2, the information needs more detailed elaboration; sources 77, 83, 93-97, 101 are outdated, need to be checked

In response to the suggestion, Section 2.2 has been revised to provide additional details on LRRK2, particularly focusing on its metabolism. The sources (77, 83, 93-97, and 101) have been updated and cited with the most recent studies available on the subject.

8

Reference 146, 170, 194, 195, needs clarification, no title of the work.

As per the suggestion, references 146, 170, 194, and 195 have been replaced and cited with the most relevant and recent studies on the subject in all instances.

9

There are many outdated references in the work, 173, 177, 178, 187, 194, 195, etc. more up-to-date information needs to be used.

In response to the feedback, references that appeared to be outdated, including 173, 177, 178, 187, 194, and 195, have been replaced and cited with the most recent studies available on the subject.

Furthermore, a thorough check of the manuscript has been conducted, and older citations have been replaced with recent ones wherever possible.

10

The review contains an excessive number of diagrams showing well-known processes associated with the pathogenesis of PD. It is recommended to reduce the number of schemes and make them more general.

In response to the reviewer’s valuable suggestion above, the figures in the manuscript have been increased to enhance the detailed overview of the subject, aiming to improve understanding and clarity of the concepts discussed. Schematic illustrations play a crucial role in a review paper from the reader’s perspective. If the reviewer has specific recommendations for removing any particular figure, we are open to those suggestions and will gladly make the necessary adjustments.

11

Stylistic elaboration of the text is necessary; in many places the text of the review article looks raw and unedited.

Thanks for pointing this out. The language, in terms of grammar, sentence structure, and stylistic errors, has been checked and improved accordingly at the necessary instances in the entire manuscript.

12

English is not clear everywhere. Significant stylistic correction is required.

Thank you for your feedback. We have carefully reviewed and improved the language in terms of grammar, sentence structures, and stylistic errors to enhance clarity throughout the manuscript.

Reviewer 3 Report

Comments and Suggestions for Authors

Dear Authors, "Advancements in Genetic and Biochemical Insights: Unraveling the Etiopathogenesis of Neurodegeneration in Parkinson’s Disease” is a great written review article.

The article is well-researched and provides a comprehensive overview of the subject matter.

Especially the structured presentation of the content, makes it easy for readers to follow the key points. The inclusion of nice overview figures greatly enhances the clarity of the concepts discussed. However, I believe a slightly more detailed description for Figure 1 would be beneficial for readers to fully grasp its significance.

I observed that out of the 328 citations, a substantial 177 are from publications within the last 10 years, indicating the relevance and up-to-dateness of the information presented. Nevertheless, I noticed some minor inconsistencies in the use of abbreviations, specifically in the case of "SH-SY5Y," where the hyphen is inconsistently applied. To enhance reader comprehension, I recommend a thorough review to ensure uniformity in the application of abbreviations throughout the document. Additionally, considering the multitude of abbreviations used, it might be advantageous to include an appendix with an abbreviation list to facilitate reader understanding.

Furthermore, I noted a minor typographical error in line 394, where an extra space is present, and in line 609, where an extra closing parenthesis is included.

While I appreciate the thorough exploration of the central nervous system in your article, I would like to suggest the inclusion of discussions on the enteric nervous system, the gut-brain axis, and the role of the microbiome. These factors play a crucial role in the broader context and would further enrich the comprehensiveness of your review.

Finally, I would like to express once again my respect for the high quality of your work. The proposed improvements are aimed at enhancing the overall reading experience and ensuring clarity for your readers.

Comments on the Quality of English Language

only minor spelling errors have to be corrected

Author Response

Responses to Reviewer 3 Comments

S. No.

Comments

Responses

REVIEWER 3

1

Especially the structured presentation of the content, makes it easy for readers to follow the key points. The inclusion of nice overview figures greatly enhances the clarity of the concepts discussed.

However, I believe a slightly more detailed description for Figure 1 would be beneficial for readers to fully grasp its significance.

Thank you for your valuable feedback and positive comments on the structured presentation and clarity of the content. We appreciate your acknowledgment of the nice overview figures and their role in enhancing the understanding of the concepts.

Regarding your suggestion for a slightly more detailed description of Figure 1, a detailed overview of Figure has been included in its caption.

2

I observed that out of the 328 citations, a substantial 177 are from publications within the last 10 years, indicating the relevance and up-to-dateness of the information presented.

Nevertheless, I noticed some minor inconsistencies in the use of abbreviations, specifically in the case of “SH-SY5Y,” where the hyphen is inconsistently applied. To enhance reader comprehension, I recommend a thorough review to ensure uniformity in the application of abbreviations throughout the document.

Additionally, considering the multitude of abbreviations used, it might be advantageous to include an appendix with an abbreviation list to facilitate reader understanding.

Thank you for your observation and remarks. The abbreviation “SH-SY5Y” has been addressed for its inconsistent use and corrected in all instances.

All the abbreviations used in the entire manuscript have been thoroughly reviewed and addressed, ensuring their uniform application throughout the document.

3

Furthermore, I noted a minor typographical error in line 394, where an extra space is present, and in line 609, where an extra closing parenthesis is included.

Suggested corrections have been made.

4

While I appreciate the thorough exploration of the central nervous system in your article, I would like to suggest the inclusion of discussions on the enteric nervous system, the gut-brain axis, and the role of the microbiome. These factors play a crucial role in the broader context and would further enrich the comprehensiveness of your review.

Thanks for your appreciation and suggestion.

The enteric nervous system, the gut-brain axis, and the role of the microbiome in PD pathogenesis have been included in the manuscript (lines- 111-131).

Round 2

Reviewer 2 Report

Comments and Suggestions for Authors

The article has been significantly improved by the authors and can be recommended for publication in Biomolecules

Comments on the Quality of English Language

The quality of English is satisfactory